# State-dependent coupling of hippocampal oscillations

**Brijesh Modi[1]\*[§][†], Matteo Guardamagna[2][‡], Federico Stella[2], Marilena Griguoli[1,3], Enrico Cherubini[1], Francesco P Battaglia[2]\***

[1]European Brain Research Institute, Rome, Italy; [2]Donders Institute for Brain, Cognition and Behavior, Radboud University, Nijmegen, Netherlands; [3]CNR, Institute of Molecular Biology and Pathology, Rome, Italy

**Abstract** Oscillations occurring simultaneously in a given area represent a physiological unit of brain states. They allow for temporal segmentation of spikes and support distinct behaviors. To establish how multiple oscillatory components co-vary simultaneously and influence neuronal firing during sleep and wakefulness in mice, we describe a multivariate analytical framework for constructing the state space of hippocampal oscillations. Examining the co-occurrence patterns of oscillations on the state space, across species, uncovered the presence of network constraints and distinct set of cross-frequency interactions during wakefulness compared to sleep. We demonstrated how the state space can be used as a canvas to map the neural firing and found that distinct neurons during navigation were tuned to different sets of simultaneously occurring oscillations during sleep. This multivariate analytical framework provides a window to move beyond classical bivariate pipelines for investigating oscillations and neuronal firing, thereby allowing to factor-in the complexity of oscillation–population interactions.

**\*For correspondence:**
brijeshmodi12@gmail.com (BM);
F.Battaglia@science.ru.nl (FPB)

**Present address:** [†]Norwegian Centre for Molecular Medicine, University of Oslo, Oslo, Norway; [‡]Kavli Institute for Systems Neuroscience and Centre for Neural Computation, Norwegian University of Science and Technology, Trondheim, Norway

[§]Lead Contact

**Competing interest:** The authors declare that no competing interests exist.

## Editor's evaluation

Traditional approaches for the study of brain oscillations are typically based on analyzing spectral features of individual oscillations (univariate methods) or the power and phase relationship between two oscillations (bivariate methods). This manuscript presents a different, multivariate, approach to simultaneously analyze interactions between multiple oscillations and applied it to rodent hippocampal LFPs. This innovative and important method provides a comprehensive and convincing approach to characterize oscillatory network states, opening new avenues for studying the complex interactions that characterize neural circuit dynamics.

## Introduction

Oscillations, generated by large neuronal ensembles, are a hallmark of the mammalian brain. They are well preserved during evolution (*Buzsáki and Draguhn, 2004*) and have been suggested to play a key role in high cognitive functions (*Uhlhaas and Singer, 2010*). They enable the synchronization of neural activity within and across brain regions, thus promoting the precise temporal coordination of neural processes underlying memory, perception, and behavior (*Colgin and Moser, 2010*). Their disruption leads to cognitive and sensorimotor deficits associated with several neuropsychiatric diseases (*Uhlhaas et al., 2011*). Oscillations occurring at different frequency bands (from 0.5 to 200 Hz), interact with each other through hierarchical cross-frequency coupling (*González et al., 2020*; *Jensen and Colgin, 2007*; *Tort et al., 2008*). Such interactions are linked to different computational operations or different phases in a computational operation and are thought to characterize distinct brain states supporting various aspects of behavior (*McCormick et al., 2020*).

In the hippocampus, theta cycles are often nested with bouts of faster oscillations in the gamma frequency and are thought to be related to different stages of memory processing (*Lopes-Dos-Santos et al., 2018*). Specifically, during wakefulness, the CA1 region is characterized by cross-frequency coupling between theta and distinct gamma oscillations, which exhibit their peak power at distinct phases of ongoing slow oscillations (*Belluscio et al., 2012*; *Buzsáki and Wang, 2012*; *Colgin et al., 2009*; *Csicsvari et al., 2003*; *Scheffer-Teixeira et al., 2012*; *Yamamoto et al., 2014*). The slow, medium, and fast gamma oscillations, emerging from *stratum radiatum (SR)*, *stratum lacunosum moleculare (SLM)*, and *stratum pyramidale (SP)*, respectively, are linked to network activity in entorhinal cortex, CA3 and CA1 region, respectively (*Colgin et al., 2009*; *Fernández-Ruiz et al., 2017*; *Guardamagna et al., 2023*; *Lasztóczi and Klausberger, 2016*; *Schomburg et al., 2014*). During sleep, theta and medium gamma oscillations dominate the rapid eye movement REM sleep, whereas delta, beta frequency band, and ripples or fast gamma (100–200 Hz), linked to memory consolidation processes, dominate the non-REM sleep (*Battaglia et al., 2011*; *Buzsáki, 1989*; *Genzel et al., 2014*; *Roumis and Frank, 2015*). The composition of hippocampal states by distinct set of slow and fast oscillations reflects synchrony among different brain areas and corresponds to distinct functional states of the hippocampal network (*Colgin, 2015*). However, despite extensive research on simultaneously occurring oscillations, very little is known about the dynamic variations in the composition of hippocampal network state with time and behavior.

Oscillations are generated by neuronal population, and they in turn are known to modulate the firing activity of individual cells (*Benchenane et al., 2010*; *Hulse et al., 2017*; *O'Keefe and Recce, 1993*). Classically, neuronal firing has been characterized in the context of a single-frequency band during distinct brain states (*Fox et al., 1986*; *Henze et al., 2000*; *Hafting et al., 2008*; *van Wingerden et al., 2010*; *Siapas et al., 2005*). However, the classical analytical methods, due to their bivariate nature, are limited to examining the firing activity of neurons in relation to power or phase or frequency of a specific oscillation only, thus neglecting how multiple oscillations, occurring simultaneously in a given region, modulate the neuronal firing in a combinatorial manner.

Here, a novel analytical approach has been described to investigate how simultaneously occurring network oscillations dynamically contribute to the composition of hippocampal states and how they influence the neuronal firing. To this aim, multisite local field potentials (LFPs), recorded from the CA1 region of the dorsal hippocampus of freely moving mice, were used to construct the network state space during wakefulness and sleep. This method has allowed to create a compact representation of the state of multiple oscillatory processes, distinct in frequency and anatomical localization. We used this compact representation for studying various oscillations, their intrinsic organization, their temporal progression, and their simultaneous influence on distinct neuronal ensembles and behavior. In addition, the state space provided a window for observing the hippocampal population in the context of the network in which they are embedded. This allowed us to examine how cells fire as a function of the network state and how multiple oscillations simultaneously modulate cell firing. As a proof of concept, this approach has been applied to datasets recorded, in similar experimental conditions, from another animal species (rats), in Dr. Gyorgy Buzsaki's lab at NYU (*hc-11 dataset*, crncs.org; *Grosmark et al., 2016*; *Grosmark and Buzsáki, 2016*). Lastly, the network state space framework has been applied to study alterations in the organization of hippocampal oscillations in mice lacking neuroligin 3 (NLG3 KO; neuroligin-3 knock out), an animal model of autism (*Baudouin et al., 2012*).

## Results
### Experimental paradigm and construction of the network state space
On a single day, we recorded brain activity in freely moving mice (n = 4) during four consecutive trials: (1) Sleep1, (2) Novel Arena-1 Exploration, (3) Novel Arena-2 Exploration, and (4) Sleep2, (*Figure 1A*). Layer-resolved LFP were extracted from the CA1 region of the dorsal hippocampus using multisite silicon probes (*Figure 1B*; *Guardamagna et al., 2022*). LFPs from SP, SR, and SLM layers were used to extract signals in the following frequency bands: delta (1–5 Hz), theta (6–10 Hz), beta (10–20 Hz), slow gamma (20–45 Hz), medium gamma (60–90 Hz), and fast gamma (100–200 Hz). Each of these 18 frequency bands (six from each layer, *Figure 1C*) were used to compute median power in nonoverlapping bins of 200 milliseconds and then smoothed using Gaussian kernel. The resulting 18 power time series across all four trials combined (divided into, say, N bins of 200 ms each) form a cloud of

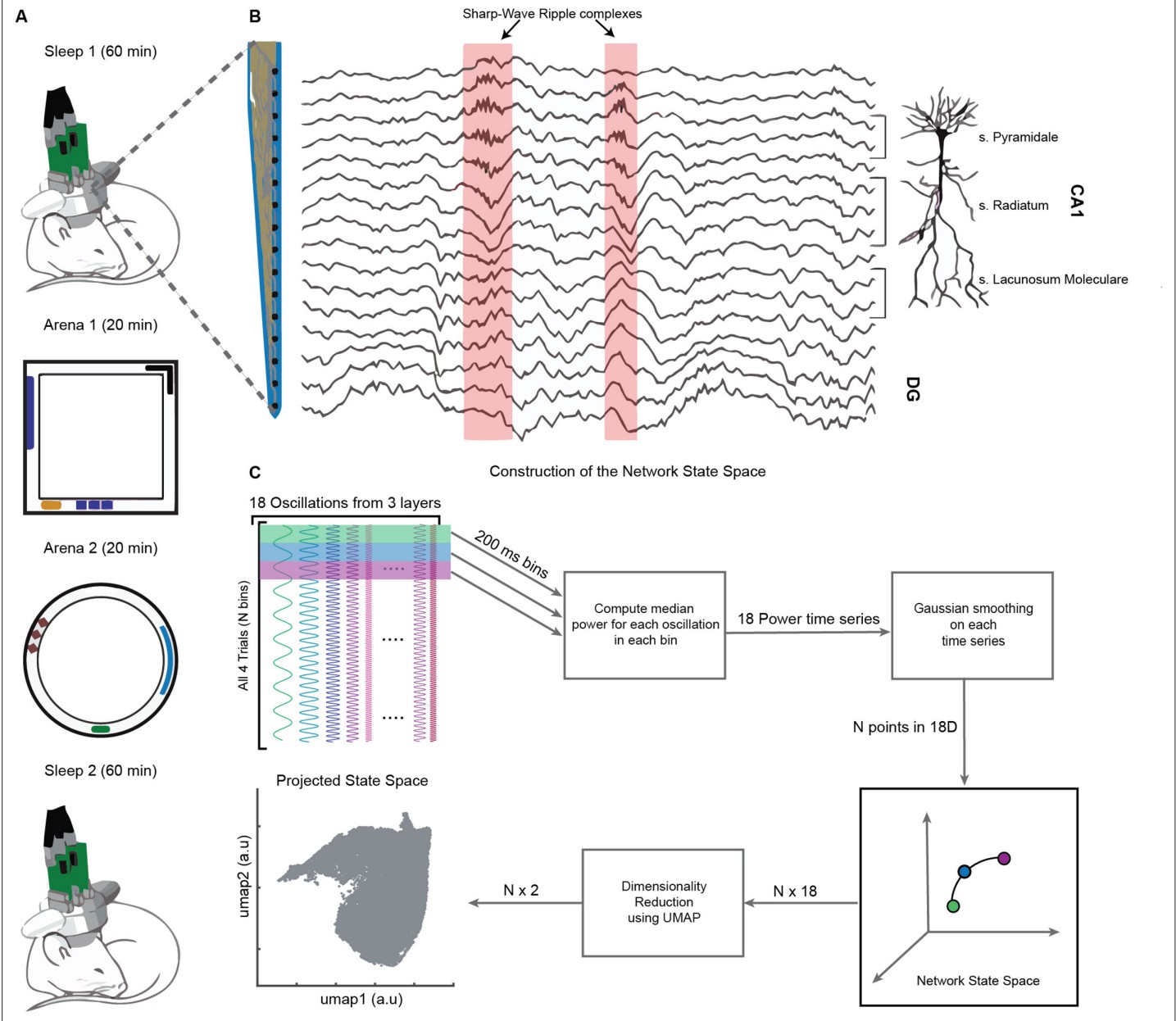

**Figure 1.** Experimental paradigm and pipeline for construction of the network state space. (**A**) Representative four-trial sequence for recording during sleep and awake exploration (top to bottom). (**B**) Representative traces of local field potential (LFP) recorded from various layers of dorsal CA1 using silicon probe. (**C**) Pipeline for the construction of network state space. See also *Figure 1—figure supplements 1 and 2*.

The online version of this article includes the following figure supplement(s) for figure 1:

**Figure supplement 1.** Positions of tetrodes targeting dorsal CA1 in freely moving mice.

**Figure supplement 2.** Accelerometer signal in arbitrary units overlaid on network state space used for the identification of sleep states during rest periods.

N points in 18D space, which we refer to as the network state space (*Figure 2A*). Each point in the cloud represents the state of the network (i.e., the power configuration of the 18 frequency bands from three layers) at a given time. Uniform Manifold Approximation and Projection (UMAP, *McInnes et al., 2020*) was employed to reduce the dimensionality of the network state space from 18D to 2D (*Figures 1C and 2A*). A similar state space was constructed using the power in current source density (CSD) signals instead of LFPs (*Figure 1—figure supplement 1*). We next characterized some fundamental properties of the network state space during sleep and wakefulness.

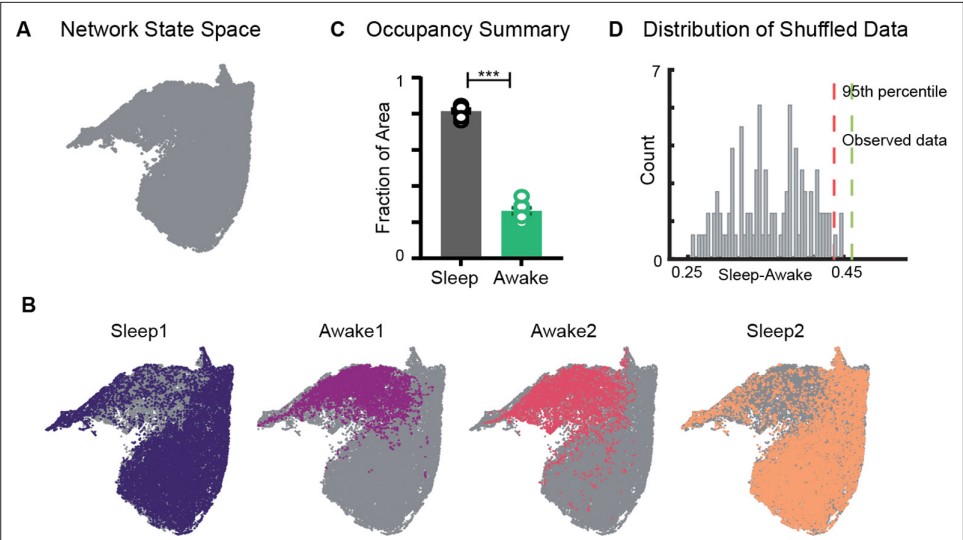

**Figure 2.** Restricted occupancy of the network state space during wakefulness. (**A**) Network state space for all four trials combined. (**B**) Trial-specific states (colored) and unvisited states (gray) on the network state space. (**C**) Fraction of state space occupied during sleep $(0.81 \pm 0.01)$ and awake $(0.26 \pm 0.01)$ trials, suggesting significant restrictions during awake trials (n = 8 sleep and awake trials from four mice. p=0.0002, Mann–Whitney test). (**D**) Distribution of difference in fraction of state space occupied during sleep and awake trials (*Sleep-Awake*) of surrogate data and observed data. See also *Figure 2—figure supplements 1–4*.

The online version of this article includes the following figure supplement(s) for figure 2:

**Figure supplement 1.** Trial-specific states overlaid on current source density (CSD) and local field potential (LFP) network state space.

**Figure supplement 2.** Network state space (local field potential [LFP]) computed with standardized features (z-scored).

**Figure supplement 3.** Restricted occupancy on the network state space in rats.

**Figure supplement 4.** Subspace occupancy visualized using principal component analysis (PCA).

## Restricted occupancy of the network state space during wakefulness

We first examined the occupancy of the network state space during sleep and awake exploration by overlaying trial specific states (states that network visits in a given trial) on the network state space (*Figure 2A and B*). We observed that during awake trials the activity of the network was restricted to a subset of the states, whereas during sleep periods, it occupied a significantly larger area of the state space. This was quantified by computing the fraction of the state space occupied during each trial. In all four mice (eight sleep and eight awake trials), we observed a significant restriction (*Figure 2C*, *Figure 2—figure supplement 1*) of the state space's occupancy during wakefulness compared to sleep $(0.26 \pm 0.01 \text{ versus } 0.81 \pm 0.01)$. The observed restriction on the state space was further compared against a surrogate data generated by randomly shuffling the binned power in all frequency bands (see 'Methods'). We found that the observed difference in occupancy between sleep and awake trials was significantly different from those computed in surrogate data (*Figure 2D*). To assess whether the observed restricted occupancy was due to smaller trial duration of awake trails (20 min) compared to sleep (60 min), we performed control analysis by computing occupancy on the state space obtained by randomly selecting equal number of network states from awake and sleep trials. We observed similar restrictions (*Figure 2—figure supplement 1C*), suggesting that the restricted occupancy during wakefulness is independent of trial duration. Next, we assessed whether the observed restrictions on the state space are driven by large fluctuations in low-frequency bands, which may weigh preponderantly in the UMAP. We computed the state space with normalized features (*Figure 2—figure supplement 2*) as well as obtained the state space projection using an alternative dimensionality reduction approach (principal component analysis; *Figure 2—figure supplement 4*). Similar patterns of restrictions on the network state space, during wakefulness, across all different projections were observed. To further validate our findings, we computed the network state space

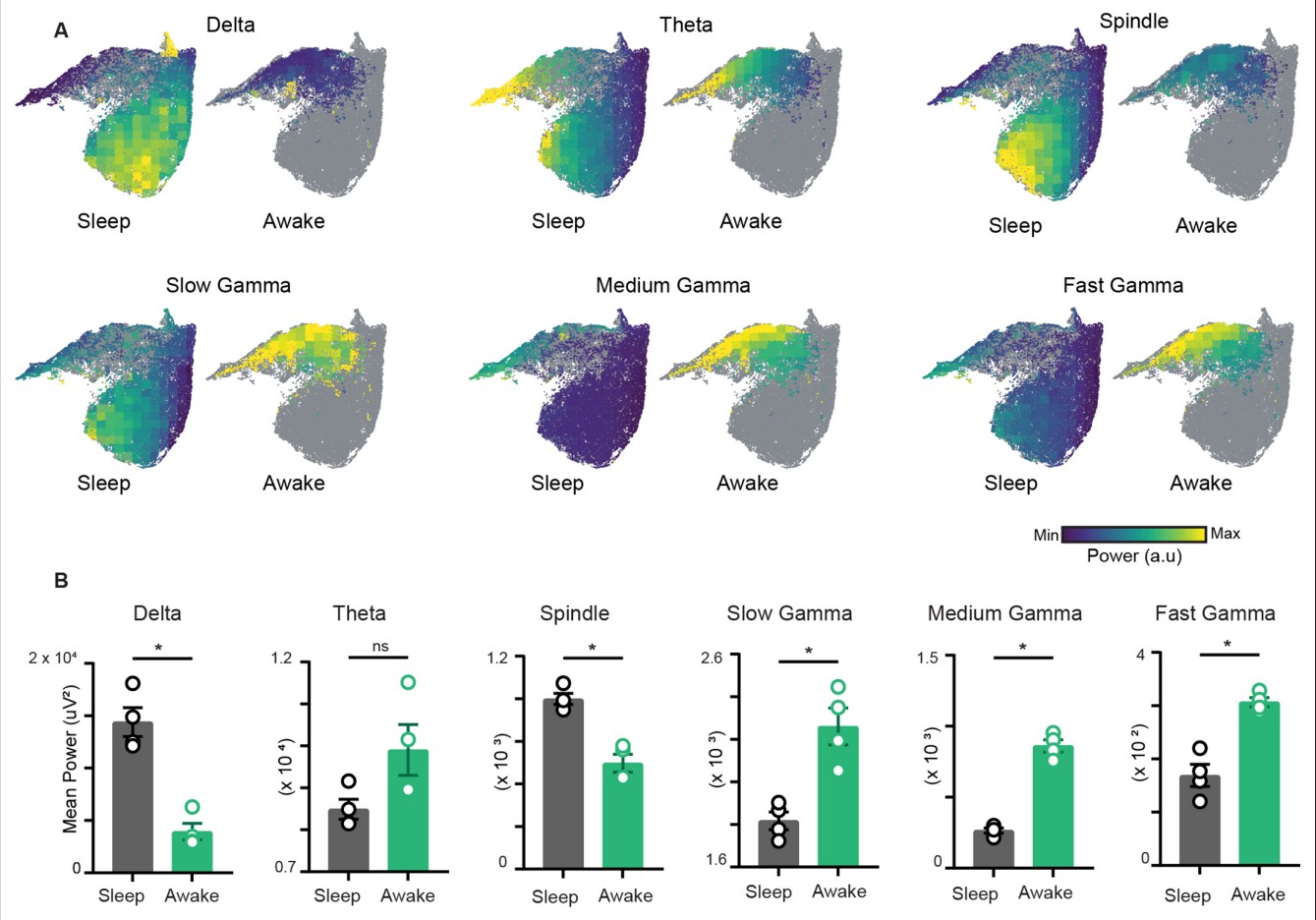

**Figure 3.** Characterization of restrictions on the network state space during wakefulness. (**A**) Distribution of power on state space for six oscillations of layer *stratum pyramidale*: delta (1–5 Hz), theta (6–10 Hz), beta (10–20 Hz), slow gamma (20–45 Hz), medium gamma (60–90 Hz), and fast gamma (100–200 Hz). Each oscillation is individually color-scaled to its respective minimum and maximum power. Unvisited states are in gray. The overlay maps demonstrate how the power of each oscillation varies on the state space and across sleep and awake trials. (**B**) Mean power comparison between awake and sleep trials using classical approach (n = 4 mice). All p<0.05, except for theta (p=0.11), Mann–Whitney test.

and occupancy for the datasets recorded from freely moving rats in similar experimental conditions (***Figure 2—figure supplement 3***) and observed remarkable similarities in restrictions on the state space, suggesting that functional organization of network oscillations during sleep and wakefulness is conserved across species. In addition, we observed that the network occupies the same set of states when animals explore distinct arenas (***Figure 2B***, ***Figure 2—figure supplement 1***). Hence, the restrictions are primarily dependent on the behavioral state of the animal rather than environmental factors.

## Characterization of restrictions on the network state space during wakefulness

To characterize the network restrictions and visualize how different hippocampal rhythms vary simultaneously during sleep and awake trials, we overlaid the power of each oscillation on the network state space (***Figure 3A***). While classical analytical approaches employ bivariate methods by evaluating each oscillation individually across sleep and awake trials (***Figure 3B***), the network state space employs multivariate approach and highlights how fluctuations in power for one oscillation link to the general oscillatory state of the network. This allows visualizing how the power of one oscillation varies in the context of all other oscillations. In addition, it underscores the regime of operation for different oscillations during sleep and wakefulness. For instance, the restricted subspace of the state space occupied during wakefulness corresponds to states with low delta power, moderate beta power, and moderate to high gamma power. However, during sleep, network oscillations exhibit broader range

of operations since the power of each oscillation varies from its minima to its maxima as evident in the overlay maps (*Figure 3A*). This characteristic distribution during sleep was further used to determine the localization of REM, non-REM, and intermediate sleep states on the network state space.

## Distinct localization of REM and non-REM sleep states on state space

Sleep trials were further classified into REM, non-REM, and intermediate states using classical definitions (see 'Methods'). We overlaid the *theta/delta* power ratio for each state on the network state space (*Figure 4A*). States with high theta/delta ratio were identified as REM (*Figure 4A, C and D*, *Figure 4—figure supplement 1*). Similarly, non-REM sleep states were detected by overlaying *delta × beta* power product for each state on the state space (*Figure 4B–D*). The remaining states were classified as intermediate sleep states. These overlay maps allowed to visualize how functionally distinct sleep states are localized on the state space and they were further used to study how network properties varies in distinct regions of the state space. Visual inspection of power overlay maps (*Figure 3A* and REM and non-REM states localization maps in *Figure 4A and B*) revealed that REM states are characterized by moderate to high power in beta, slow, medium, and fast gamma oscillatory bands simultaneously with theta oscillations, whereas non-REM states are characterized by the presence of moderate to high power in theta, slow, and fast gamma oscillations along with delta and beta. Notably, the medium gamma oscillations operate in low-power mode during non-REM sleep. This co-occurrence of network oscillations is formally quantified in later sections.

## Increased density of non-REM states associated with theta and gamma oscillations after exploration

We next computed network density that represents the fraction of time the network spends in each bin on the network state space (*Figure 4E*). It is measured in the units: *number of state visits/bin/second*. Density on the state space computed during awake trials measures the normalized occurrence of corresponding behaviors during exploration whereas density computed during sleep trials measures the time spent by the network in various sleep states (REM, non-REM). Mirroring the smaller occupied state space region, the median density of awake trials was significantly higher than that detected during sleep (0.13 ± 0.009 versus 0.03 ± 0.0005). We next investigated the density across sleep trials (Sleep1 and Sleep2) by comparing the median density of REM and non-REM states between the two sleep trials (*Figure 4F*). We observed an increase in the median density of non-REM states during post-exploration sleep (*Figure 4G*: center; Sleep1: 0.039 ± 0.001versus Sleep2: 0.048 ± 0.002) while the density of REM sleep remained identical (*Figure 4C*, left; Sleep1: $0.0076 \pm 5e - 4$ versus Sleep2: 0.0075 ± 0.001). In particular, non-REM states in Sleep2 tended to concentrate in a region of increased power in the delta and beta bands, which could be the results of increased interactions with cortical activity modulated in the same range. It is also likely that such effect was induced by the exposure to relevant behavioral experience. In fact, changes in density of individual oscillations after learning have been reported using traditional analytical methods and are thought to support memory consolidation (*Bakker et al., 2015*; *Eschenko et al., 2008*; *Eschenko et al., 2006*). Nevertheless, while traditional methods provide information about individual components, the novel approach used here provides additional information about the combinatorial shift in the dynamics of network oscillations after learning or exploration. Thus, it provides the basis for identifying how coordinated activity among different oscillations supports memory consolidation processes as those occurring during non-REM sleep after exploration, which cannot be elucidated using traditional analytical methods.

In this study, although REM and non-REM sleep states were identified using classical methods, the state space allowed quantifying changes in density of all other oscillations such as theta, slow, medium, and fast gamma that occur simultaneously with delta and beta bands (from power distribution maps [*Figure 3A*] and REM and non-REM maps [*Figure 4A*]). Thus, identification of sleep states on state space provides additional information about oscillation-specific changes across sleep trials. However, whether other oscillations may be statistically related to delta and beta during non-REM states of sleep and whether such correlation depends on the state of the network remain to be elucidated.

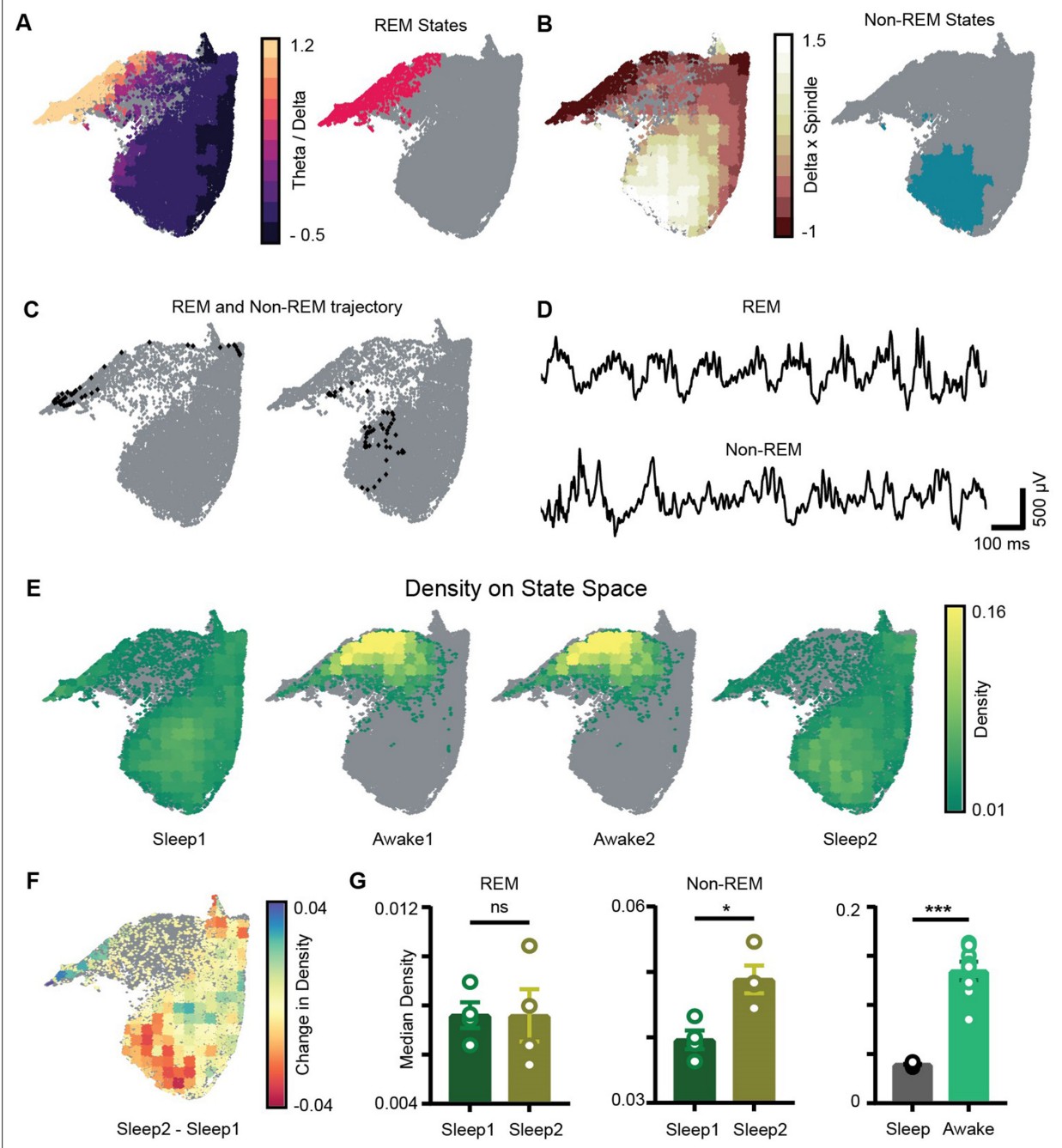

**Figure 4.** Localization of rapid eye movement (REM) and non-REM sleep states and density on the network state space. (**A**) Left: standardized theta/delta ratio from pyramidal layer overlaid on state space. Right: identified REM states (magenta colored). (**B**) Left: standardized delta × beta product from pyramidal layer overlaid on state space. Right: identified non-REM states (cyan colored). (**C**) Representative REM and non-REM states on the network state space. (**D**) Corresponding REM and non-REM local field potential (LFP) from pyramidal layer. (**E**) Density overlaid on state space across all four trials (unvisited states are colored gray). (**F**) Representative change in density map across two sleep trials (Sleep2 – Sleep1). (**G**) Left: comparison of median REM density between two sleep trials (n = 4 mice); (Sleep1: $0.0076 \pm 5e-4$ v/s Sleep2: 0.0075 ± 0.001, p=0.88 Mann–Whitney test); Center: comparison of median non-REM density between two sleep trials (n = 4 mice); (Sleep1: 0.039 ± 0.001 v/s Sleep2: 0.048 ± 0.002, p<0.05, Mann–Whitney test); right: median density comparison between and sleep and awake trials (n = 8 sleep and awake trials, four mice) (0.03 ± 0.0005 v/s 0.13 ± 0.009, p=0.0002 Mann–Whitney test). See also *Figure 4—figure supplement 1*.

The online version of this article includes the following figure supplement(s) for figure 4:

**Figure supplement 1.** Trajectories on the network state space and raw local field potential (LFP) from CA1 pyramidal layer.

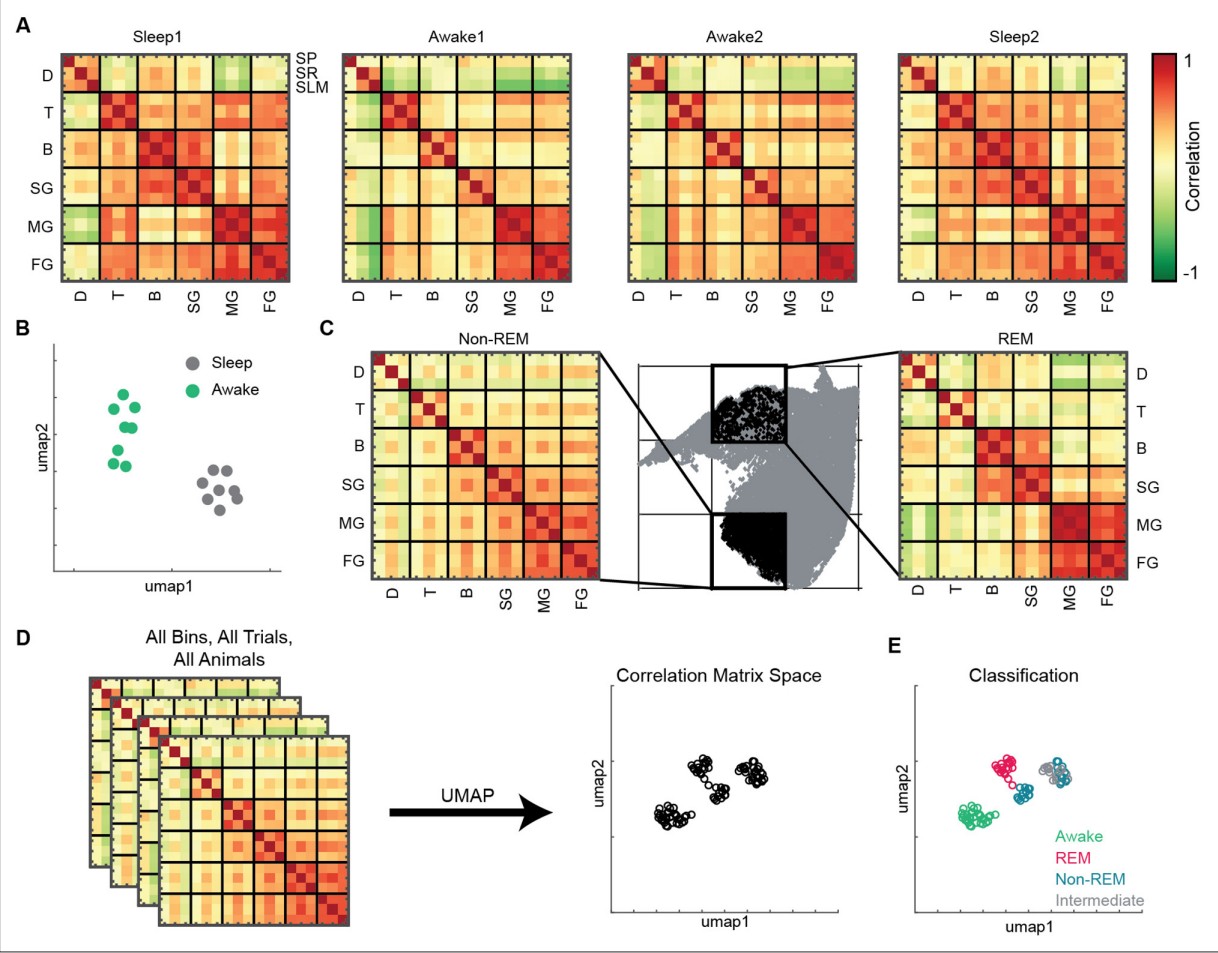

**Figure 5.** State and layer-dependent coupling of hippocampal oscillations. (**A**) Representative pairwise correlation among 18 oscillations (D, delta; T, theta; B, beta; SG, slow gamma; MG, medium gamma; FG, fast gamma) from three layers (stratum pyramidale [SP], stratum radiatum [SR], stratum lacunosum moleculare [SLM]). Rows are arranged by combining all three layers for each oscillation. (**B**) Correlation matrix space generated by using matrices in (**A**) from all the mice (total 16 points, 8 sleep and 8 awake trials from four mice). Each point represents the correlation matrix of oscillations. The separation of sleep and awake points in the correlation matrix space suggests distinct nature of coupling among oscillations during sleep and wakefulness. (**C**) Binned state space (gray) for a sleep trial. Representative rapid eye movement (REM) and non-REM bins highlighted (black) and their corresponding correlation matrix of oscillations. (**D**) Correlation matrix space constructed using matrices from all bins collected from all trials and all animals. Each point on correlation matrix space projection represents a correlation matrix. (**E**) Bin status (awake, REM, non-REM, intermediate sleep) overlaid on correlation matrix space, exhibiting state dependent coupling of hippocampal oscillations ($p < 0.0001$ using multivariate ANOVA).

## State and layer-dependent coupling of hippocampal oscillations

To investigate state-dependent coupling among various oscillatory processes, we computed pairwise correlation matrix using binned power in all 18 frequency bands (see 'Methods'). Each correlation matrix represents how various bands are coupled during sleep and awake trials (*Figure 5A*). We observed distinct configuration of frequency band's correlations during sleep and awake trails (*Figure 5B*). We further investigated how coupling of bands varies on the state space by binning the state space (*Figure 5C*) and computed correlation matrix in each bin. We classified the bins into one of the following states: awake, REM, non-REM, and intermediate sleep (see 'Methods'). We employed UMAP to visualize the variability across correlation matrices. (*Figure 5D*) and observed distinct coupling configurations for awake, REM, and non-REM states (*Figure 5E*). During REM, we found increased coupling (positive correlation) among beta and slow gamma oscillations as well as among medium and fast gamma bands accompanied by decoupling (negative correlation) with delta. This organization among oscillations was altered during non-REM sleep as all oscillations except delta were coupled. Gamma segregation and delta decoupling offer a picture of hippocampal REM sleep as being more akin to awake locomotion (with the major difference of a

stronger medium gamma presence) while also suggesting a substantial independence from cortical slow oscillations. On the other hand, the across-scale coherence of non-REM sleep is consistent with this sleep stage being dominated by brain-wide collective fluctuations engaging oscillations at every range. Distinct cross-frequency coupling among various individual pairs of oscillations such as theta-gamma, delta-gamma, etc., have been already reported (*Bandarabadi et al., 2019*; *Clemens et al., 2009*; *Hammer et al., 2021*; *Scheffzük et al., 2011*). However, computing cross-frequency coupling on the state space provides the additional information on how multiple oscillations, obtained from distinct CA1 hippocampal layers (*stratum pyramidale, stratum radiatum, and stratum lacunosum moleculare*), are coupled with each other during distinct states of sleep and wakefulness. Furthermore, projecting the correlation matrices on 2D plane provides a compact tool that allows to visualize the cross-frequency interactions among various hippocampal oscillations. Altogether, this approach reveals the complex nature of coupling dynamics occurring in hippocampus during distinct behavioral states.

After characterizing the static properties of state space such as occupancy, power distribution, localization of sleep states, network state density, and coupling among oscillations, in the following analyses we addressed the dynamic properties of state space such as network flow, state transitions, and speed on state space.

## Alterations in sleep state transitions after exploration/learning

To investigate how learning during exploration affects the consequent sleep, we characterized state transition patterns during Sleep1 and Sleep2. By plotting outgoing trajectories on the network state space (*Figure 6A*, *Figure 6—figure supplement 1*), we visualized state transitions during sleep. We quantified the probability of state transitions among various sleep states (REM, non-REM, etc.) in a transition matrix (*Figure 6B*). We calculated changes in transition probabilities across sleep trials by measuring average absolute change in probability (AACP, see 'Methods'). This allowed to quantify the amount of sleep state transitions altered after exploration/learning. To assess whether these changes in transition probabilities of sleep states are random or not, we generated 1000 pairs of sleep and awake trials by randomly shuffling state transitions from sleep data and computed transition matrices, difference matrix, and their corresponding AACP. The AACP value of real data was then compared to the distribution of randomly shuffled trials. We observed that the AACP value of real data was beyond the 95th percentile of the distribution of shuffled data (*Figure 6D*), Altogether, this data highlights the specific alterations in general sleep architecture and hippocampal oscillatory landscape following learning/novel exploration.

## Intra-state sleep transitions are more plastic than inter-state transitions

We further examined which transitions on the state space are significantly altered across sleep trials. We computed AACP specifically for transition from REM/non-REM/intermediate sleep state to REM/non-REM/intermediate state. We found that transitions occurring from REM-to-REM sleep and non-REM-to-non-REM sleep (intra-state transitions) are more vulnerable to plasticity after exploration as compared to inter-state transitions (such as non-REM to REM, REM-to-intermediate, etc.) (*Figure 6E and F*). These changes in intra-state transitions were observed to be beyond randomness (*Figure 6—figure supplement 2A and B*), indicating a specificity in plastic changes in state transitions after exploration. In particular, while the average REM period duration is unaltered after exploration (*Figure 4G*), REM temporal structure is reorganized. In fact, increased probability of REM to REM transitions indicates a significant prolongation of REM bout duration. Similarly, the increase in non-REM to non-REM transition probability reflects an increased duration of non-REM bouts. Therefore, environment exploration was accompanied by an increased separation between REM and non-REM periods, possibly as a response to increased computational demands. More in general, the network state space allows to characterize the state transitions in hippocampus and how they are affected by novel experience or learning. By observing the state transition patterns, this analytical framework allows to detect and identify state-specific changes in the hippocampal oscillatory dynamics, beyond the possibilities offered by more traditional univariate and bivariate methods. We next investigated how fast the network flows on the state space and assessed whether the speed is uniform or whether it exhibits specific region-dependent characteristics.

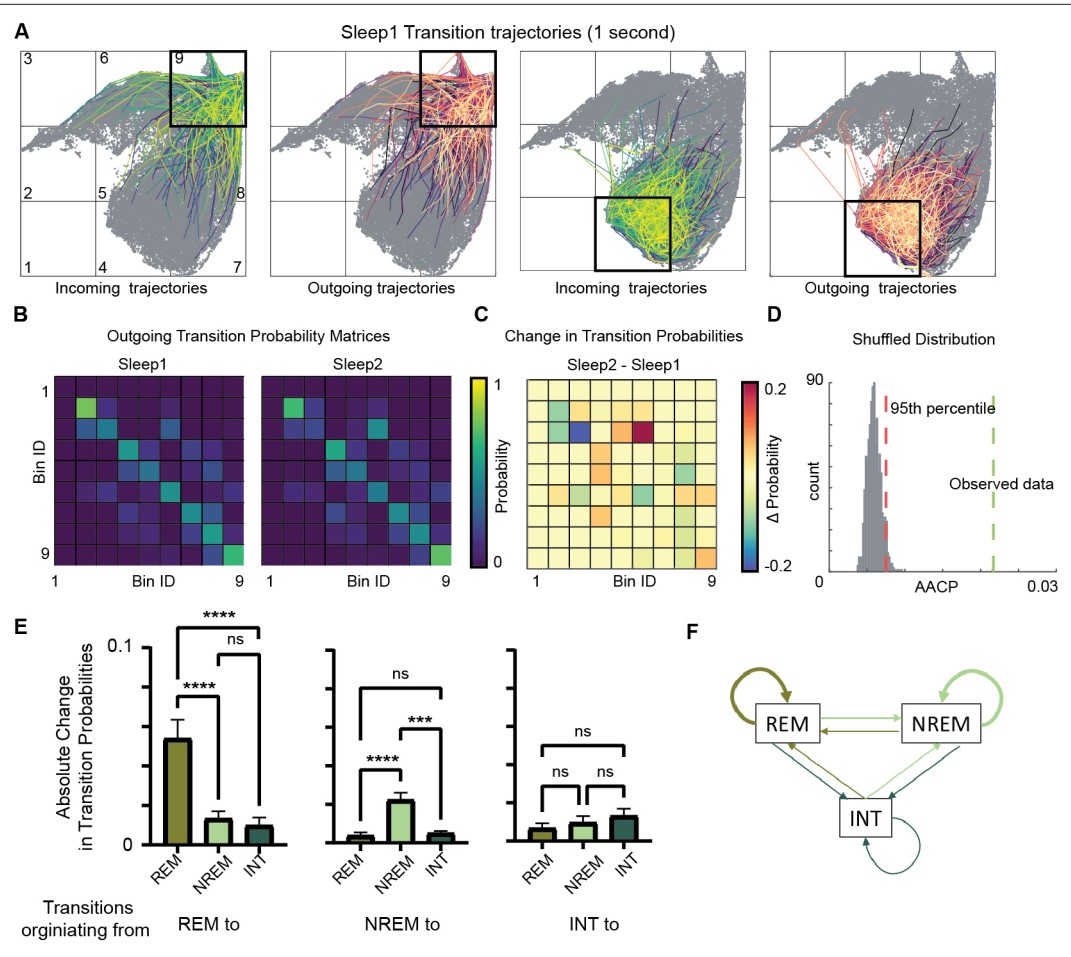

**Figure 6.** Alterations in sleep state transitions after exploration. (**A**) Incoming and outgoing trajectories for two representative bins on state space during a sleep trial. Each trajectory spans 1 s in time before (incoming) and after (outgoing) the occurrence of a given state in each bin. (**B**) State transition matrices for two sleep trials: pre-exploration sleep (Sleep1) and post-exploration sleep (Sleep2). (**C**) Change in transition probabilities across two sleep trials obtained by subtracting two transition matrices (Sleep2 – Sleep1). (**D**) Comparison of observed sleep data's average absolute change in probability with shuffled sleep data. Average absolute change in probability (AACP) across sleep trials for observed data = 0.021, 95th percentile of shuffled data = 0.007. (**E**) AACP across sleep trials for transitions originating from rapid eye movement (REM), non-REM, and intermediate (INT) state to REM, non-REM, and intermediate sleep state (REM: p<0.0001; non-REM: p<0.0001; INT: p=0.38, one-way ANOVA). (**F**) Schematic diagram of absolute change in transition probabilities across sleep trials. Arrow's thickness corresponds to absolute change in transition probabilities. See also *Figure 6—figure supplements 1 and 2*.

The online version of this article includes the following figure supplement(s) for figure 6:

**Figure supplement 1.** Outgoing trajectories (1 s in future) for all bin of a representative sleep trial.

**Figure supplement 2.** Comparison of average absolute change in probability (AACP) of transition matrices with shuffle data.

## Stabilization on state space during transition to REM

Coverage speed determines how fast the network sweeps the area of the state space. To calculate the coverage speed, we binned the state space as shown in *Figure 7A and B* and computed the number of bins the network covers in each time window (see 'Methods'). Thus, the resultant time series provides information about how network's flow varies over time. When speed is closer to its minimum value (i.e., 1 bin/s), the network stabilizes to a set of states that are closer to each other, whereas when speed approaches its theoretical maximum value, the network covers wider range of area, thus more potential fluctuations in the power of oscillations. We observed a reduction in median coverage speed during awake compared to sleep trials (*Figure 7C*, *Figure 7—figure supplement 1*).

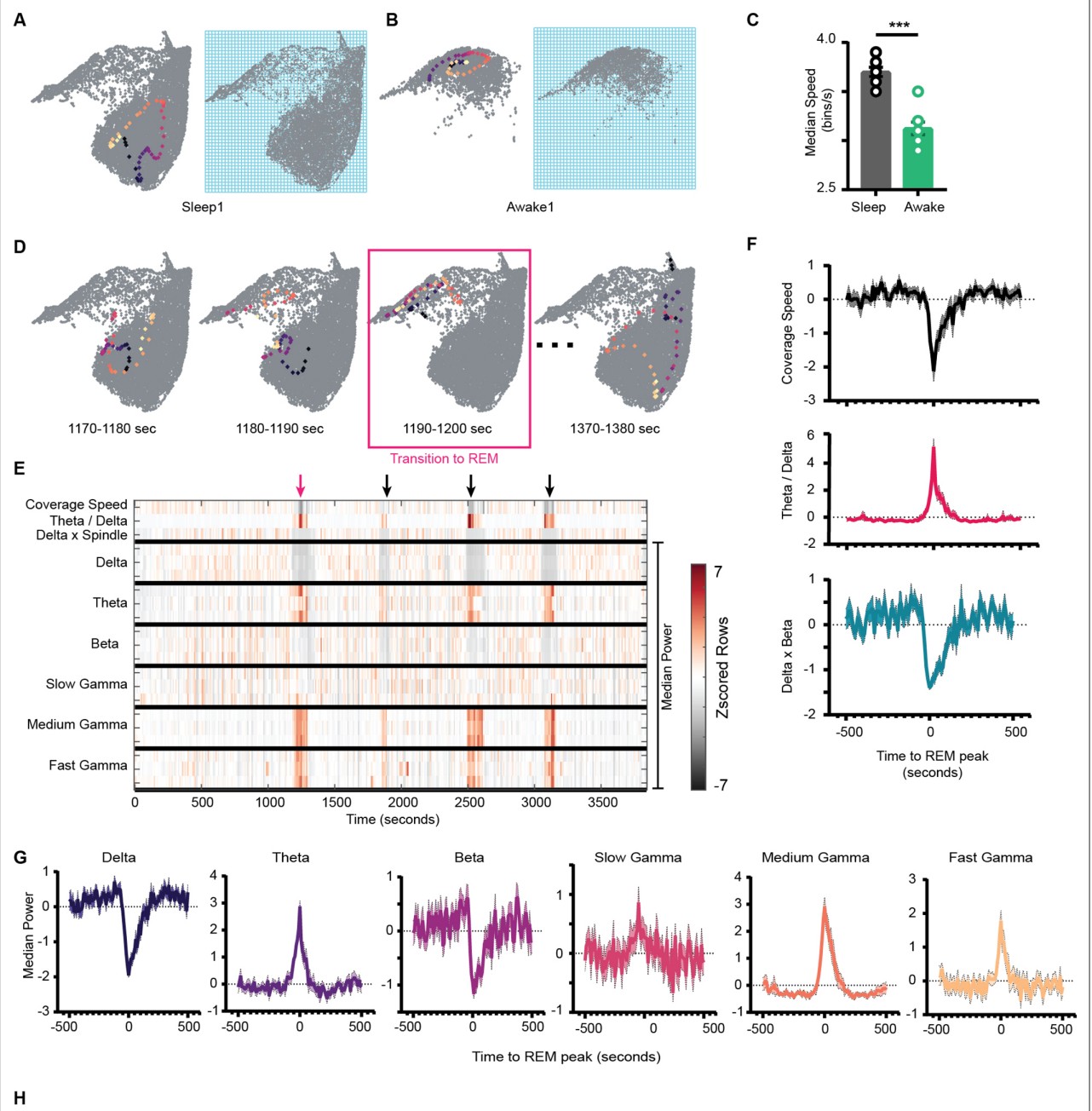

**Figure 7.** Network stabilization during transition to rapid eye movement (REM). (**A**) Left: representative trajectory in 10 s time frame on state space during sleep trial (trajectory starts from blue to yellow). Right: binned state space used to compute the speed. (**B**) Same as (**A**), but for awake trial. (**C**) Median speed for sleep and awake trials (n = 8 awake and sleep trials from four animals) (Sleep: 3.7 ± 0.04 v/s Awake: 3.12 ± 0.06 bins/s), p=0.0003, Mann–Whitney test. (**D**) Representative trajectories in 10 s time frame before (panels 1 and 2) network transitions to REM state (panel 3, red box), and when network exits REM (panel 4). (**E**) Median power v/s time for entire sleep trial with 18 oscillations, speed of coverage, theta/delta ratio, and delta × beta product. Arrows indicate network transition to REM. Red arrow corresponds to example shown in (**D**). Each row is independently z-scored. Three rows for given oscillation corresponds to three layers (*stratum pyramidale, radiatum, and lacounosum-moleculare*, respectively). Note the changes in median power of oscillations as network transition to REM. (**F**) Summary plot for 19 REM bouts collected across eight sleep trials from four mice. Top: speed of coverage (units: bins/s) center: theta/delta ratio bottom: delta × beta power product showing dip in coverage speed as network transitions to REM. All plots are standardized using z-score. (**G**) Average power for six oscillations from pyramidal layer for 19 REM bouts as network transitions to REM. All are statistically significant at T = 0 s when compared with T = –500 s except slow gamma, p<0.05, Mann–Whitney test. See also *Figure 7— figure supplement 1*.

The online version of this article includes the following figure supplement(s) for figure 7:

**Figure supplement 1.** Speed of coverage on normalized state space.

During awake trials, due to subspace occupancy and behavioral constraints (such as walking, running, stopping, exploring walls, etc.) on state space, the network operates in constrained space compared to sleep. Thus, the network is more likely to fall on states closer to each other, reducing its coverage speed compared to sleep trials. We further explored the coverage speed during sleep, wondering how coverage speed varies during network transitions to REM sleep. We addressed this issue by taking a closer look at network trajectories and coverage speed when the network makes the transition (*Figure 7D*). We observed a sharp dip in coverage speed associated with state transition to REM sleep (*Figure 7F*, S10A). This further suggests that during the transition to REM sleep the network stabilizes on state space.

### Dichotomous nature of gamma during transition to REM

We next investigated the characteristics of stabilization during transition to REM. We wondered how the power in frequency bands is reorganized during the transition. We investigated this by plotting a spectrogram of median power for all 18 frequency bands during the entire sleep trial (*Figure 7E*). This allowed to visualize how speed of coverage varies with median power in all oscillatory bands. By definition of REM, we observed an increase in theta and a reduction in delta power during the transition. However, this transition was also associated with a significant reduction of power in beta frequency band and with an increase of medium and fast gamma power (*Figure 7G*). While slow gamma remained mostly unchanged during the REM epoch, we measured a significant average increase in its power in a window preceding the onset of REM sleep and extending for few tens of seconds. This result is consistent with previous reports of slow gamma power decreasing over the initial part of a REM epoch (*Zhang et al., 2019*).

In the following section, we explored two applications of this analytical framework. We first used the network state space as a canvas (or context) to map the firing (activation signatures) of cells from CA1 pyramidal layer.

### Distinct sleep state signatures for cells with different firing rates and sparsity during awake exploration

We visualized the network modulation of CA1 cells by overlaying cell's firing rate on network state space. These plots represent how each cell activates as a function of network states (*Figure 8B*, *Figure 8—figure supplement 1B*). They allow to visualize how distinct oscillations simultaneously co-vary with the firing of individual cells during wakefulness and sleep. Each cell has its characteristic firing pattern on the state space (activation signature), as it does in the arena during exploration (firing rate map) (*Figure 8A*, *Figure 8—figure supplement 1A*). We hypothesized that distinct activation signatures on state space may correspond to distinct firing rate maps during exploration. To investigate this, we projected sleep signatures of all recorded cells for a given animal on a 2D plane (see 'Methods'). We refer to this projected space as *signature space* (*Figure 8C*) as each point on signature space represents a cell's activation signature during sleep trials. We then overlaid mean firing rate (*Figure 8D*, *Figure 8—figure supplement 2A*) and mean sparsity (which measures spatial specificity) (*Figure 8E*, *Figure 8—figure supplement 2B*) of cells, as computed during awake trials, on signature space. We observed distinction of sleep signatures for low firing cells in the arena compared to the high-firing ones (*Figure 8F*; see also *Figure 8—figure supplement 3* for interneurons and pyramidal cells). Specifically, we observed that sparse firing cells during exploration exhibit preferential firing to non-REM states (from *Figures 4A, B and 8F*) and are therefore simultaneously modulated by high power in delta, theta, beta oscillations, as well as moderate power in slow and fast gamma oscillations (from *Figure 3A*). However, the high-firing cells during exploration are differently orchestrated by simultaneously occurring sleep oscillations as evident in *Figure 8F*. Altogether, this suggests that distinct functional cells in hippocampal CA1 are tuned to distinct set of network states and have distinct activation signatures during sleep trials. The network may employ such switches (distinct network states) to precisely manipulate the firing of different cells, allowing them to play state-dependent functional roles. Studying cell's firing patterns on network state space (context of the network) may provide new insights into how cells are functionally embedded in a larger network.

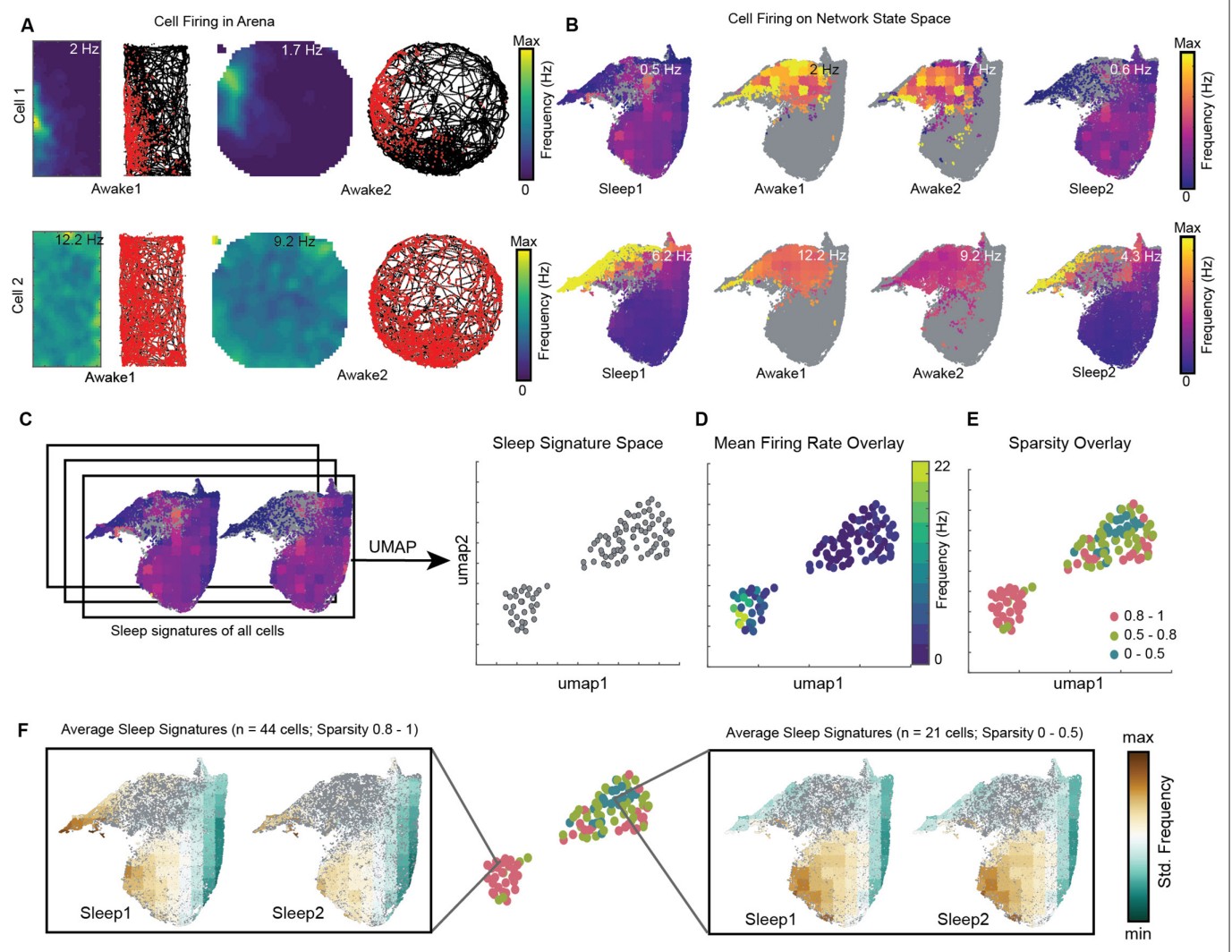

**Figure 8.** Distinct sleep state signatures for cells with different firing rates and sparsity. (**A**) Two representative cells and their firing during awake exploration trials (rectangle and circular arena). Panels 1 and 3 in both rows: firing rate map for two trials. Panels 2 and 4: animal's trajectory during exploration (black) and overlaid with spikes (red). (**B**) Firing of cells across four trials overlaid on network state space. Unvisited states are colored gray. Note that the distinct cell firing patterns in arenas correspond to distinct cell firing patterns on network state space. (**C**) Sleep signatures on state space are used to create signature space with 106 cells from a given animal. Each point represents firing signature of cell on state space across two sleep trials. (**D, E**) Mean firing rate and mean sparsity as observed during two awake trials, overlaid on sleep signature space. Note that most high-firing cells on the opposite end of the spectrum from low-firing cells suggesting distinct sleep signature for cells having distinct firing rates and sparsity during awake exploration. (**F**) Average sleep signatures for two sets of cells having low (0–0.5) and high sparsity (0.8–1) during awake exploration. See also *Figure 8— figure supplements 1–5*.

The online version of this article includes the following figure supplement(s) for figure 8:

**Figure supplement 1.** Cell firing overlaid on arenas during exploration and on the network state space.

**Figure supplement 2.** Distinct sleep signature corresponds to cells with distinct firing patterns in arena.

**Figure supplement 3.** Distinct sleep signature corresponds to distinct cell types in hippocampus.

**Figure supplement 4.** Altered organization of hippocampal oscillations in NLG3 KO mice.

**Figure supplement 5.** Characterization of state space in WT and NLG3 KO mice.

## Altered organization of oscillations in the NLG3 knock-out mouse, an animal model of autism

We next applied the network state space approach to study the organization of network oscillations in the hippocampus of the NLG3 knock-out (KO) mouse, an animal model of autism spectrum disorders

(ASD). NLGs are postsynaptic adhesion molecules that bind to their presynaptic partners neurexins to functionally couple the postsynaptic densities with the transmitter release machinery, thus contributing to synapses organization and stabilization (*Südhof, 2008*). Mice carrying the human mutations/deletions of the *Nlg3* gene are useful animal models to study the mechanisms of ASD since they recapitulate most of behavioral deficits found in autistic children. Previous studies from NLG3 KO mice have highlighted alterations in synaptic signaling and network oscillations, associated with social deficits (*Földy et al., 2013*; *Modi et al., 2019*; *Tabuchi et al., 2007*). Here, LFPs were used to extract network oscillations occurring at different frequencies from the CA1 region of awake head-restrained NLG3 KO mice and wild-type controls (WT) (*Figure 8—figure supplement 4*).

We first constructed the composite network state space of hippocampal oscillations by pooling network states from all trials and all animals (*Figure 8—figure supplement 4C*, see 'Methods,' n = 7 trials from four WT and five trials from three KO mice). Each trial consisted of 3 min of LFP recordings while the animal was awake and at rest on a platform in a head-fixed condition (*Figure 8—figure supplement 4A and B*). Assessing the network state during rest allowed us to control for any genotype specific disparity in locomotion (*Jaramillo et al., 2014*; *Modi et al., 2019*) and its associated hippocampal oscillations. By overlaying trial-specific states on the combined network state space (*Figure 8—figure supplement 4C*), we observed distinct occupation of state space by WT mice compared to NLG3 KO mice. Overlay maps of power of different oscillations on the state space suggest that mice lacking NLG3 occupy state space associated with lower power of oscillations (*Figure 8—figure supplement 4D*, top). Comparing power of individual bands reveals significant reduction in mean power of delta, theta, beta, medium, and fast gamma frequency bands. Only slow gamma power was not significant. We next computed correlation among various oscillatory processes (*Figure 8—figure supplement 5A and B*) and observed no differences across genotypes. Lastly, we computed the speed of coverage and observed no difference across genotypes. Altogether, this suggests altered co-occurrence patterns along with intact coupling and transition properties of hippocampal oscillations in NLG3 KO mice (*Figure 8—figure supplement 5C*).

## Discussion

In this study, we introduced an analytical framework for investigating the organization of hippocampal oscillations. This framework is established by constructing a state space of multiple network variables: in our case, the power of 18 hippocampal frequency bands from three layers of the CA1 region. While we used predefined frequency bands to filter oscillations from LFPs, alternatively, unsupervised methods (see *Lopes-Dos-Santos et al., 2018*) can be employed to extract oscillations in an unbiased manner and construct the state space of network oscillations. We applied this framework to characterize various static and dynamic properties of the network state space during wakefulness and sleep and demonstrated multiple applications of the framework.

Traditional approaches usually consist of analyzing spectral features of individual oscillations (univariate methods) or the power and phase relationship between two oscillations (bivariate methods). This multivariate analytical approach provides a unique window to assess the combinatorial effect of more than two oscillations on the state of the network, the neuronal population and the behavior. This provides a formal framework to uncover the potential combinations of neural signals underlying various cognitive functions during sleep and wakefulness. By constructing the state space of power in various frequency bands from distinct layers of a brain region, our analysis allows to characterize the network landscape and how various oscillatory processes co-vary as a function of the overall state of the network. Additionally, we demonstrated that the state space can be used as a canvas to map the activity of individual cells, thus allowing to study how various oscillations can simultaneously influence the firing of neuronal ensembles. Although the scope of this study was limited to demonstrate the state space approach for power of network oscillations, the general idea can be extended to create a state space with additional attributes of oscillations such as instantaneous frequency, phase, etc. The resulting state space may provide additional insights about the organization of network oscillations. For instance, characterization of the state space constructed with instantaneous phase of the oscillations may provide novel information on how cell spiking is phase-locked to multiple oscillations simultaneously and how those phase-locking properties vary with time and behavioral state of an animal.

We first used the network state space to visualize the restricted nature of access to the state space during wakefulness compared to sleep. We then characterized that restriction by overlaying

the power of individual oscillations on the state space. This helped us to visualize how oscillations are intrinsically organized and co-vary simultaneously on the state space during wakefulness and sleep. Compared to sleep, in our experimental settings, during awake trials, the delta power is low, and the network is more dominated by theta and gamma supporting the ongoing navigational needs (*Buzsáki and Moser, 2013*; *Colgin and Moser, 2010*). Interestingly, it was recently reported (*Furtunato et al., 2020*) that a specific increase in delta power accompanied by a decrease in theta and gamma power in rats trained to execute 35 successive short-term treadmill run at the same speed. These results suggest, compared to our experiments, exploration of a larger subspace, at least in part related to changes in internal physiological states during performance of running exercise. We also applied the analysis to the datasets obtained from rat's hippocampus and observed remarkable similarity across species (*Figure 2—figure supplement 3*), suggesting that restrictions on network state space during wakefulness are conserved across species. These findings are in agreement with previous reports characterizing multiregion state space (*Gervasoni et al., 2004*). Together, these observations indicate that, during wakefulness, the hippocampal network occupies a fraction of subspace, independently of task and behavioral needs.

We next identified REM and non-REM states on the network state space. Alternatively, they can also be identified using oscillations from multiple brain areas (*Bagur et al., 2018*). We observed increased non-REM density during post-exploration sleep trials. This finding complements a previous report (*Eschenko et al., 2008*) of sustained increased in sharp wave ripples activity during slow wave sleep after learning. In our case, the animal explores two novel environments during exploration trials, subsequently in the post-exploration sleep, the network spends more time on the non-REM states, possibly supporting the occurrence of sharp wave ripples and memory consolidation. Along with increase in the occurrence of high delta and high beta power states (definition of non-REM), we have also observed simultaneous increase in the occurrence of other network oscillations such as theta, slow, and fast gamma in post-exploration sleep trial (Sleep2). The slow and fast gamma oscillations are linked to inputs from CA3 and medial entorhinal cortex, respectively (*Colgin et al., 2009*; *Fernández-Ruiz et al., 2017*), thus an enhanced density of states characterized by medium and fast gamma suggests an increased hippocampal-entorhinal and CA1-CA3 communication during non-REM sleep after exploration or learning. This modulation of active oscillatory states across sleep stages is also consistent with observed reorganization of cellular excitation between REM and non-REM sleep (*Grosmark et al., 2012*; *Miyawaki and Diba, 2016*), which might mirror the differential inputs impinging on the network. Nevertheless, the precise mechanisms and functional role(s) of such increased inter-areal communication after exploration or learning remain to be determined.

Further, we characterized the coupling of various frequency bands during awake and sleep trials and across distinct network states. We reported state-dependent coupling of distinct hippocampal oscillations, suggesting that their sources change their functional link during distinct states on state space. For instance, during REM sleep, we observed beta-frequency range oscillations coupled with slow gamma power and medium gamma coupled with fast gamma. We also observed delta decoupled from medium and fast gamma. These observations are in agreement with a general understanding of REM sleep as corresponding to a state of low coherence across the cortex and between the cortex and the hippocampus (*Diekelmann and Born, 2010*) while arguing for the existence of frequency-specific channels of communication. In fact, this configuration alters during other network states such as non-REM sleep (*Figure 5C*). Besides the modulation of power of individual oscillations, the observed variations in coupling across frequency bands amplitudes might indicate a wider spectrum of combinations available to the network to dynamically regulate the activity of hippocampal population. Furthermore, such coupling states could mirror a similarly complex processing of specific inputs from up or downstream regions during distinct states of sleep and wakefulness (*Colgin, 2015*; *Dvorak et al., 2021*).

We next investigated the state transitions on the network state space. We quantified trajectories on state space and observed that state transition probabilities vary between two sleep trials and two awake trials. During exploration trials, the flow of the hippocampal network state is strongly affected by ongoing spatial navigation. The incoming and outgoing state space trajectories are in fact bounded by the details of the exploration activity statistics (such as speed). Changes in transition probabilities thus reflect alterations in behavioral sequences exhibited by the animal during exploration. During sleep trials, the state of the hippocampal network flows without any interference from

environmental inputs. The incoming and outgoing trajectories bounds are instead characteristics of specific sleep states. Thus, changes in transition probabilities across sleep trials represent modifications in the probable paths on the network state space. These changes might be due to learning and/or experience acquired during awake exploration trials. The experience of running on a track or in an arena may change the state of the circuit in many ways: (1) by changing the neuromodulatory state due to arousal, physical exertion; (2) by changing brain temperature, which is known to modify brain potentials (*Moser et al., 1993*); and (3) by imprinting a trace of the experience by synaptic plasticity. Our methods enable to capture theses changes in terms of the coordinated modifications in oscillatory phenomena. Further experiments will be needed to disentangle possible causes of these changes and unveil how these dynamical changes correlate with modifications in information processing in the hippocampus.

We then characterized the speed of coverage on the network state space and reported the stabilization of network during transitions to REM sleep. We also described how frequency bands are reorganized during the transition. We observed specific increase in power of medium and fast gamma, but not slow gamma, during the transition to REM. State-dependent cross-frequency coupling between theta and gamma was also reported during REM sleep in EEG recordings from mouse parietal cortex (*Scheffzük et al., 2011*). These results suggest the presence of common organizing principles across cortical and subcortical areas. It is worth mentioning that the slow gamma activity, which is not affected by the transition to REM, originates not only in the CA3 region of the hippocampus but also in the CA2 (*Alexander et al., 2018*; *Middleton and McHugh, 2016*). CA2 shows distinct anatomical (*Hitti and Siegelbaum, 2014*; *Kohara et al., 2014*) and functional (*Dudek et al., 2016*) properties that may differently affect REM sleep-dependent information processing.

Further, we demonstrated two applications of this analytical method. Firstly, rather than using individual oscillations to study the firing properties of hippocampal neurons, we overlaid the firing of hippocampal neurons on state space and used those sleep signatures to distinguish cells in CA1. We reported a gradient in the sleep state space signatures of cells, differentiating cells with sparse activation in the arena, from more active ones. This suggests that the sparse firing cells in the hippocampus are modulated differently by network oscillations occurring simultaneously during sleep.

Secondly, this method was applied to study the organization of hippocampal network state space in mice lacking NLG3, animal models of autism, which exhibit social deficits reminiscent of those found in autistic children (*Bariselli et al., 2018*; *Modi et al., 2019*; *Radyushkin et al., 2009*). In line with data obtained from animal models of ASD (*Hammer et al., 2015*; *Modi et al., 2019*; *Paterno et al., 2021*) or from the EEG/MEG of autistic children (*Cornew et al., 2012*; *Larrain-Valenzuela et al., 2017*; *Ortiz-Mantilla et al., 2019*; *Rojas and Wilson, 2014*; *Wang et al., 2013*), we observed a significant reduction in delta, theta, beta, medium, and fast gamma activity in NLG3 KO mice. These data are similar to those reported in the CA2 region of the hippocampus of NLG3 KO anaesthetized animals (*Modi et al., 2019*), suggesting that alterations in network activity in CA2 may influence the downstream CA1 network activity (*Hitti and Siegelbaum, 2014*; *Kohara et al., 2014*). However, the reduction in slow gamma power detected in CA2 (*Modi et al., 2019*) was not associated to a similar decrease in CA1 reported here, suggesting slow gamma oscillations in CA1 can emerge from sources other than CA2, possibly CA3 (*Colgin et al., 2009*).

Oscillatory activities, and in particular the gamma ones, depend on the synchronous firing of large neuronal ensembles of excitatory cells within and across brain regions, which are paced by GABAergic interneurons (*Gonzalez-Burgos and Lewis, 2008*). Therefore, the observed alterations in power may be due to a reduced GABA release from GABAergic interneurons such as those containing parvalbumin (PV+) and cholecystokinin (CCK+) known to contribute to generate theta and gamma rhythms (*Klausberger et al., 2005*; *Tukker et al., 2007*). Interestingly, in a previous study from NLG3 KO mice (*Földy et al., 2013*), a clear increase in GABA release from CCK+GABAergic interneurons into CA1 principal cells was detected due to the impairment of tonic endocannabinoid signaling. Additionally, an impairment of GABAergic currents was also reported in CA2 region of NLG3 KO mice (*Modi et al., 2019*), an upstream region to CA1 (*Hitti and Siegelbaum, 2014*; *Kohara et al., 2014*). Therefore, the reduced oscillatory activity reported here is probably mediated by the loss of $GABA_A$-mediated neurotransmission in both CA1 and CA2 hippocampal regions. This would cause an enhancement of network excitability with consequent excitatory/inhibitory unbalance, critical for controlling spike rate and information processing.

Lastly, we highlight some other potential applications of the network state space framework:

*(i) Assessment of inter-areal communication.* The state space approach can be applied to study oscillations from multiple brain regions simultaneously and how one set of oscillations from one region influence the others (*Gervasoni et al., 2004*). *(ii) Analysis of EEG datasets.* Various oscillations can be extracted from EEG recorded from multiple brain areas and used to assess how they are organized during various behavioral tasks during wakefulness (i.e., decision-making, navigation, imagination, mental calculation) or during sleep. Additionally, this analytical framework can be applied to study alterations in the organization of brain-wide network oscillations as signatures or biomarkers in various neurodevelopmental and neurodegenerative disorders. *(iii) Network assessment of novel therapeutics.* Newly developed drugs are required to be extensively tested for their harmful side effects. Using network state space method will allow assessing how multiple brain regions, and their oscillations are simultaneously affected under pharmacological treatment. Additionally, this information can be used to predict potential side effects and develop better therapeutics in drug development.

## Methods

### Animals

#### Recordings from freely moving mice

Four male C57BL/6J mice (Charles River) were used in this study, all implanted with a Hybrid Drive. All animals received the implant between 12 and 16 wk of age. After surgical implantation, mice were individually housed on a 12 hr light-dark cycle and tested during the light period. Water and food were available ad libitum.

#### Recordings from head-restrained mice

Experiments were performed on offspring male derived from heterozygous mating after 10 backcrossing with C57BL/6J. Results were analyzed blindly before genotyping. Control experiments were performed on wild-type littermates and C57BL/6J (WT). Genotyping was carried out on tail biopsy DNA by PCR using a standard protocol. After surgical implantation, mice were individually housed on a 12 hr light-dark cycle and tested during the light period. Water and food were available ad libitum.

#### Recordings from freely moving rats

Datasets of extracellular recordings from right dorsal hippocampus of three Long-Evans rats using silicon probes were used for validating the findings. Detailed information is published in *Grosmark and Buzsáki, 2016*.

### Surgical procedures and data acquisition

#### Extracellular recordings from freely moving mice

The fabrication of the Hybrid Drives and the implantation surgeries were done as described earlier (*Guardamagna et al., 2022*). These experiments were performed in Nijmegen, Netherlands. From post-surgery day 3 onward, animals were brought to the recording room and electrophysiological signals were investigated during a rest session in the home cage. Each day tetrodes were individually lowered in 45/60 μm steps (1/4 of a screw turn) until common physiological markers for the hippocampus were discernible (SWR complexes during sleep or theta during locomotion). Silicon probe signals were used as additional depth markers. Electrophysiological data were recorded with an Open Ephys acquisition board (*Siegle et al., 2017*). Signals were referenced to ground, filtered between 1 and 7500 Hz, multiplexed, and digitized at 30 kHz on the head stages (RHD2132, Intan Technologies, USA). Digital signals were transmitted over two custom 12-wire cables (CZ 1187, Cooner Wire, USA) that were counter-balanced with a custom system of pulleys and weights. Waveform extraction and automatic clustering were performed using Dataman (https://github.com/wonkoderverstaendige/dataman; *Eichler, 2020*) and Klustakwik (*Rossant et al., 2016*), respectively. Clustered units were verified manually using the MClust toolbox. Manual waveform curation was performed using the MClust software suite. Identified CA1 units were not further classified into pyramidal cells and interneurons. During all experiments in freely moving animals, video data was recorded using a CMOS video camera (Flea3 FL3-U3-13S2C-CS, Point Grey Research, Canada; 30 Hz frame rate) mounted above the arena.

Animal position data was extracted offline using a deep learning-based software, Deeplabcut (*Mathis et al., 2018*).

## Extracellular recordings from head-fixed mice

Separate set of mice were used for awake head-fixed recordings at EBRI, Rome, Italy. Seven male mice (four WT and three NLG3 KO) aged 3–5 mo were implanted with stainless steel head bar (Luigs & Neuman, Germany). Mice were individually housed after implantation and were allowed to recover for 1 wk. Mice were progressively habituated to head-fixation on a horizontal platform until they no longer attempt to escape (3–4 wk). For craniotomy, they were anesthetized with i.p. injection of a mixture of tiletamine/zolazepam (Zoletil; 80 mg/kg) and xylazine (Rompun, 10 mg/kg), a day before the recording. A glass electrode (Hingelberg, Malsfeld, Germany) with the resistance of 1–2 MΩ, filled with standard Ringer's solution containing (in mM) 35 NaCl, 5.4 KCl, 5 HEPES, 1.8 $CaCl_2$, and 1 $MgCl_2$, was lowered to target CA1 pyramidal layer using micromanipulator (Scientifica, UK). Location of electrode was assessed by visually inspecting LFP trace, depth information, and was confirmed post hoc using brain sectioning and immunohistochemistry. LFPs were acquired with Multiclamp 700B amplifier (Molecular Devices, USA) and digitized with an A/D converter (Digidata 1550, Molecular Devices). Data were acquired at sampling rate of 10 kHz and were further downsampled to 1 kHz using MATLAB for all the further analysis of state space. Multiple 3 min trials were recorded from a mouse, and artifact-free trials were considered for all further analyses.

## Behavioral paradigm

### In datasets recorded from mice

In the open-field experiments, mice were free to explore two arenas for about 20 min each. The arenas were either square (45 cm × 45 cm) or circle (45 cm diameter) or rectangle (50 cm × 25 cm). This was preceded and followed by a rest session in the animal's home cage (Sleep1 and Sleep2) that typically lasted 60 min each.

### In datasets recorded from rats

Experiments on rats were performed in Buzsaki lab at NYU. Each session consisted of a long (~4 hr) PRE rest/sleep (Sleep1) epoch home cage recordings performed in a familiar room, followed by a Novel MAZE running epoch (~45 min) in which the animals were transferred to a novel room and water-rewarded to run on a novel maze. These mazes were either (1) a wooden 1.6 m linear form, (2) a wooden 1 m diameter circular platform, or (3) a 2 m metal linear platform. Animals were rewarded either at both ends of the linear platform or at a predetermined location on the circular platform. The animal was gently encouraged to run unidirectionally on the circular platform. After the MAZE epochs, the animals were transferred back to their home cage in the familiar room where a long (~4 hr) POST rest/sleep (Sleep2) was recorded. Detailed paradigm and recording conditions are described in *Grosmark and Buzsáki, 2016*.

### Histology

After the final recording day, tetrodes were not moved. Animals were administered an overdose of pentobarbital (300 mg/ml) before being transcardially perfused with 0.9% saline, followed by 4% paraformaldehyde solution. Brains were extracted and stored in 4% paraformaldehyde for 24 hr. Then, brains were transferred into 30% sucrose solution until sinking. Finally, brains were quickly frozen, cut into coronal sections with a cryostat (30 microns), mounted on glass slides, and stained with cresyl violet. The location of the tetrode tips was confirmed from stained sections (*Figure 1—figure supplement 1*).

## Neural data analysis

### Preprocessing of raw signals

Movement and other artifacts were removed from LFP signals by setting an amplitude threshold (~>6 median absolute deviation). Threshold were adjusted by visual inspection of raw traces. Data points with amplitude greater than threshold were eliminated from analysis pipeline. Raw LFPs recorded at 30 kHz were first low-pass filtered (<1000 Hz) and then downsampled to 1 kHz for all further analyses

of state space. Oscillations were extracted in six frequency bands: delta (1–5 Hz), theta (6–10 Hz), beta (10–20 Hz), slow gamma (20–45 Hz), medium gamma (60–90 Hz), and fast gamma (100-200) Hz from each of the three layers of CA1 using *eegfilt* from EEGLAB, an open-source toolbox in MATLAB (*Delorme and Makeig, 2004*). CSD were computed from LFP using methods described in *Lasztóczi and Klausberger, 2014*.

### Construction of network state space
Filtered LFP or CSD bands were used to compute power. Power time series were binned in 200 ms time bins to compute median power. They were further smoothed using Gaussian kernel with standard deviation of three bins. The resulting smoothened, binned, 18 power time series were given as input to UMAP to obtain 2D projection with parameters: *nearest neighbors: 25, minimum distance: 0.1, distance metric: Euclidean.*

### Combined state space from head-fixed mice
LFPs from all WT and NLG3 KO mice were normalized by scaling from 0 to 1. The scaled LFPs were then filtered to obtain the following frequency bands: delta (1–5 Hz), theta (6–10 Hz), beta (10–20 Hz), slow gamma (20–45 Hz), medium gamma (60–90 Hz), and fast gamma (100–200 Hz). The median band power was computed in bins of 200 ms for all six frequency bands. The network states from all the animals were merged and projected into 2D using UMAP with parameters as described before.

### Identification of sleep periods
3-Axis accelerometer data were used to identify periods of mobility and prolonged immobility during recordings in home cage (*Figure 1—figure supplement 2*). Raw accelerometer signal was binned in 200 ms nonoverlapping bins (same bin size as used for the construction of the state space). Mean values in each bin were used to compute moving variance over a 10 s window. A threshold of 0.01 was selected by visually inspecting the traces and applied to detect period of immobility in sleep trials. Periods with mobility during sleep trials were excluded from further analysis.

### Fraction of state space occupied
The projected state space was divided into 100 × 100 bins. The occupancy was computed by counting the number of bins visited during each trial. The visited bin count was then divided by the total number of bins in state space to get the fraction of state space occupied during each trial.

### Surrogate data for occupancy
The 18 power time series were time randomly time shifted to create 100 surrogate state spaces. UMAPs were obtained for each surrogate data and occupancy was computed. The measure *sleep-awake* occupancy was used to compare surrogate data with the observed data.

### Power overlay on state space
The projected state space was divided into 20 × 20 bins. For each trial and for each oscillation, median power in each bin was computed. The range of each oscillation was determined between minimum value and 95th percentile to prevent skewness of sequential color scale by extreme artifacts. Sequential colors were assigned to power values in the determined range to generate overlay plots (*Figure 3*). Unvisited bins in a trial were assigned gray color.

### Identification of REM and non-REM states
We adapted the methods from previous study for identification of REM and non-REM sleep (*Grosmark et al., 2012*; *Mizuseki et al., 2011*; *Mizuseki et al., 2009*). The projected state space was divided into 20 × 20 bins. For each bin on state space, we computed median theta/delta power ratio of all the network states for the identification of REM sleep states. For reproducibility, all the bins, having median theta/delta ratio greater than 80th percentile, were classified as REM. For the identification of non-REM states on state space, we computed median delta × beta power product for all the network states in a bin. All the bins, having median delta × beta product greater than 70th percentile, were classified as non-REM. These were further verified by visual inspection of LFP traces. The remaining

states were classified as intermediate sleep states without any further subclassification. The classified states were then visually inspected and cross-checked using accelerometer data and raw LFP traces. Oscillations from pyramidal layer only were used for identification.

### Density overlay on state space

The projected state space was divided into 20 × 20 bins. For each trial, we computed the number of network state visits in each bin and then divided the count by trial duration to get normalized states count, also known as density. Color assignment to density is as described above in power overlay methods.

### Correlation matrix of frequency bands

Correlation matrices for each trial were created using *corrcoef* function in MATLAB. Input to *corrcoef* were trial-specific power of 18 oscillations binned in 200 ms bins (trial-specific states). *corrcoef* computes pairwise Pearson correlation for all 18 oscillations for a given trial. For state-dependent coupling of frequency bands, the projected state space was divided into 3 × 3 bins. For each bin, a correlation matrix was generated using the corrcoef. The inputs to corrcoef were bin-specific power of all 18 oscillations (namely, all the states in that bin). To generate correlation matrix space, we took the lower triangular values of correlation matrices of all bins in all trials and used this as an input to UMAP. The embedding was generated using cosine distance as it brings similar points (in this case, correlation matrices) together, thus, allowing us to separate similar bins. Each bin was then assigned its status based on the following: (1) awake bins – bins collected from awake trials. (2) REM bins – bins from sleep trials having >10% of its network states identified as REM as described above. (3) Non-REM bins – bins from sleep trials having >10% of its network states identified as non-REM as described above. (4) Intermediate – the remaining bins from sleep trials.

### Trajectories on network state space

The projected state space was divided into 3 × 3 bins for sleep trials and 10 × 10 for awake trials. For all the network states in each bin, we show the lines (trajectories) connecting five states (1 s of time) preceding and proceeding the state. We label set of preceding states as incoming trajectories and proceeding states as outgoing trajectories.

### State transition matrix

Transition probability from, say, bin A to bin B was computed as follows: for every state in bin A, we collected outgoing trajectory states (fifth state from the current state, 1 s in future). Transition probability from A to B was then computed by fraction of total trajectory states that ended in B. Transition probabilities were then represented in the form of matrix.

### Change in transition probabilities and average absolute change

Difference matrices were computed by subtracting corresponding bin transition probabilities from two sleep or two awake trials. The mean of absolute values from difference matrices was used to generate summary plot for all animals.

### Surrogate sleep and awake trajectory trials

State transition occurring in both sleep (Sleep1 and Sleep2) and awake (Awake1 and Awake2) trials was merged to create a pool of state transitions. The sleep or awake pool was then split randomly into two surrogate sleep or awake trials, respectively. A total of 1000 randomly generated surrogate sleep or awake trials were generated to compute the change in transition probabilities between two surrogate sleep or surrogate awake trials and compute its average absolute change.

### Speed of coverage on state space

The projected state space was divided into 50 × 50 bins. We counted the number of bins visited by the network trajectory in a time frame of 10 s. Thus, the speed is measured in units: bins/s. The median values of coverage speed from all time frame of sleep and awake trails were used to plot summary.

## Speed of coverage during transition to REM

Entire sleep trial was divided into nonoverlapping time frames of 10 s. We then computed median theta/delta ratio for all the states in each time frame. REM peaks were identified by detecting peaks in median theta/delta ratio. Median power on state space from each oscillation in each time frame was also computed to generate spectrogram. Summary plots were generated by averaging power in oscillations from pyramidal layer, speed of coverage, theta/delta ratio, and delta × beta product for all REM peaks.

## Cell firing overlay on state space and in the arena

Spike trains of each cell were binned into nonoverlapping bins 200 ms (same bin size as applied during the construction of network state space). Thus, each network state gets a corresponding bin of spike train. For generating overlay maps, the projected state space was divided into 15 × 15 bins. For each bin on state space, we identified a set of network states in that bin and computed their mean firing rate from the corresponding bins in spike train. Plots demonstrating firing rate map in the arena were generated using the methods described in *Fyhn et al., 2007*.

## Firing properties overlay on cell's signature space

We computed the cell's firing rate on sleep state space (sleep signature) as above. We used these sleep signatures of all cells of a given animal as an input to UMAP with cosine distance and 10 neighbors. The resulting projected space is known as sleep signature space. We computed mean firing rate and mean sparsity of each cell during awake trials and overlaid it on sleep signature space.

## Acknowledgements

This research was supported in part by grants from the European Union's Horizon 2020 research and innovation program (MGATE, grant agreement no. 765549 to MG and FPB; BrownianReactivation grant agreement no. 840704 to FS), European Research Council (ERC) Advanced Grant 'REPLAY-DMN' (grant agreement no. 833964) to FPB; from Telethon (GGP 16083), Fondo Ordinario Enti (FOE D.M 865/2019), a collaboration agreement between the Italian National Research Council and EBRI and Del Monte Foundation grants to EC; from EMBO to BM (STF no. 8464).

The authors are grateful to Dr. Bryan Souza for his help with programming. Rafael Pedrosa and Dr. Ashley Kees for their inputs regarding experiments in head-restrained mice. Prof. Peter Scheiffele for providing NLG3 KO mice. Dr. Gyorgy Buzsaki and Dr. Andres Grosmark for providing the datasets in rats via Crcns.org. Dr. Michele Giugliano, Dr. Maurizio Mattia, Dr. Silvia Marinelli, Prof. Hannah Monyer, and Prof. Antonino Cattaneo for their valuable discussion and suggestions on the research.

## Additional information

### Funding

| Funder | Grant reference number | Author |
|---|---|---|
| European Union Horizon 2020 research and innovation program MGATE | 765549 | Francesco P Battaglia |
| European Union Horizon 2020 research and innovation | 840704 | Federico Stella |
| ERC Advanced Grant | 833964 | Francesco P Battaglia |
| Telethon | GGP16083 | Enrico Cherubini |
| Del Monte Foundation | | Enrico Cherubini |

| Funder | Grant reference number | Author |
|---|---|---|
| EMBO short term fellowship | 8464 | Brijesh Modi |
| Fondo Ordinario Enti (A collaboration agreement between the Italian National Research Council and EBRI) | FOE D.M 865/2019 | Enrico Cherubini |

The funders had no role in study design, data collection and interpretation, or the decision to submit the work for publication.

## Author contributions

Brijesh Modi, Conceptualization, Data curation, Formal analysis, Investigation, Visualization, Methodology, Writing – original draft; Matteo Guardamagna, Conceptualization, Data curation, Investigation, Visualization, Methodology, Writing – review and editing; Federico Stella, Marilena Griguoli, Conceptualization, Supervision, Writing – review and editing; Enrico Cherubini, Francesco P Battaglia, Conceptualization, Supervision, Funding acquisition, Writing – review and editing

## Author ORCIDs

Brijesh Modi https://orcid.org/0000-0002-0360-1755
Marilena Griguoli http://orcid.org/0000-0003-4067-8927
Enrico Cherubini http://orcid.org/0000-0002-1183-2772

## Ethics

In compliance with Dutch law and institutional regulations, all animal procedures concerning recordings from freely moving or sleeping mice were approved by the Central Commissie Dierproeven (CCD) and conducted in accordance with the Experiments on Animals Act (project number 2016-014 and protocol numbers 0029). All experiments from head-restrained animals were performed in accordance with the Italian Animal Welfare legislation (D.L. 26/2014) that implemented the European Committee Council Directive (2010/63 EEC) and were approved by local veterinary authorities, the EBRI ethical committee, and the Italian Ministry of Health (565/PR18). All efforts were made to minimize animal suffering and to reduce the number of animals used.

## Decision letter and Author response

Decision letter https://doi.org/10.7554/eLife.80263.sa1
Author response https://doi.org/10.7554/eLife.80263.sa2

# Additional files

## Supplementary files
• MDAR checklist

## Data availability

Datasets used in this study are available at Crcns.org (HC11 dataset) and Donders Repository (https://data.donders.ru.nl/collections/di/dcn/DSC_62002873_05_861). All codes are available made at https://github.com/brijeshmodi12/network_state_space, (copy archived at *Modi, 2022*).

The following dataset was generated:

| Author(s) | Year | Dataset title | Dataset URL | Database and Identifier |
|---|---|---|---|---|
| Guardamagna M, Battaglia FP | 2023 | Population coding dynamics and layer-resolved oscillation in the CA1 region of the hippocampus | https://doi.org/10.34973/cwhv-8e56 | Radbound University, 10.34973/cwhv-8e56 |

The following previously published dataset was used:

| Author(s) | Year | Dataset title | Dataset URL | Database and Identifier |
|---|---|---|---|---|
| Grosmark AD, Long J, Buzsáki G | 2016 | Recordings from hippocampal area CA1, PRE, during and POST novel spatial learning | http://dx.doi.org/10.6080/K0862DC5 | Collaborative Research in Computational Neuroscience, 10.6080/K0862DC5 |

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
