## [Editor Report]

Traditional approaches for the study of brain oscillations are typically based on analyzing spectral features of individual oscillations (univariate methods) or the power and phase relationship between two oscillations (bivariate methods). This manuscript presents a different, multivariate, approach to simultaneously analyze interactions between multiple oscillations and applied it to rodent hippocampal LFPs. This innovative and important method provides a comprehensive and convincing approach to characterize oscillatory network states, opening new avenues for studying the complex interactions that characterize neural circuit dynamics.

---

## [Decision Letter]

**Decision letter after peer review:**

Thank you for submitting your article "State Dependent Coupling of Hippocampal Oscillations" for consideration by *eLife*. Your article has been reviewed by 3 peer reviewers, including Antonio Fernandez-Ruiz as Reviewing Editor and Reviewer #1, and the evaluation has been overseen by Laura Colgin as the Senior Editor.

Essential revisions:

1. All reviewers found the method as an interesting novel approach with the potential to provide new insights into multivariate oscillatory coupling dynamics. However, they agree that the present manuscript fails to clearly illustrate what kinds of information this method can provide that cannot be derived from more traditional analysis and what is the physiological significance of the observations made from the state space model. This may require both performing a new analysis of the existing data and a more profound characterization and interpretation of the present results. The results presented need to be interpreted in physiological terms and, in several cases, additional quantifications are needed in addition to the qualitative descriptions provided.

2. The method relies on filtering LFPs in discrete frequency bands. The terminology needs to be clarified not to confound power in a certain frequency band with specific oscillatory events (e.g. spindles). The limitations of using power time series on predefined frequency bands rather than physiologically defined events (like. spindles or ripples) need to be discussed.

3. Analysis of single-cell spiking dynamics is not well documented which makes it difficult to properly interpret them. Were pyramidal cells and interneurons classified? Can this approach be useful to uncover more 'subtle' physiological differences that those existing between pyramidal cells and fast-spiking interneurons; or between REM and non-REM sleep stages in previous analysis? These questions can be addressed with additional descriptions and further analysis.

4. Several aspects of the analysis are not well justified or explained including the shuffling procedure for state occupancy, the choice of the number of states in Figure 6, how changes in oscillatory coupling between states were quantified, what is the relation of the state space with behavioral variables such as animal speed. In general, some parts of the result remain rather qualitative and descriptive. For those, additional quantifications and interpretations of their physiological significance need to be provided.

*Reviewer #1 (Recommendations for the authors):*

1. One important limitation of this method, that is not mentioned in the text, is its reliance on filtering LFPs in pre-defined frequency bands. While this is a common approach in the field, there is also numerous evidence that some oscillations, such as 'γ', do not occur in a restricted narrow band. Another approach would be to first extract oscillatory components in an unbiased manner and then construct the state space. These caveats and alternative possibilities should be discussed.

2. On several occasions throughout the text causal language is used without proper justification (for example line 344). Such sentences need to be reworded carefully to distinguish what are mere correlations from actual causality.

3. Although several analyses were conducted to illustrate potential applications of the method, in most cases is not clear what type of new information can be obtained that is also not possible to obtain with more traditional approaches. An example of this can be found in the last paragraph of page 8. Quantifying the power (and other features) of different oscillations across brain states is something that has been done multiple times employing traditional approaches. The authors need to do a better job in trying to illustrate which specific type of analysis can be done or what type of information can be extracted with his method beyond what traditional ones offer. The analysis in Figures 5 to 7 are good steps in that direction, that could be further elaborated at the expense of reducing the first, more descriptive, part of the results.

4. Related to the previous point, a comparison that could be more interesting than REM vs non-REM is the same sleep state before and after the novel experience. While spectral changes between REM and non-REM are so obvious that can be shown with simpler analytical approaches, one could expect more subtle changes from pre to post-sleep for the same state. This could be an opportunity to showcase the power of this approach.

5. Results regarding Figure 5 are interesting but its description is merely qualitative. For example, in lines 223-226 some statements regarding the variable coupling of oscillations are made without any quantification/ statistics to back them up. Those should be provided either in the main text or figure legend.

6. The language describing Figure 6 is too vague and makes it difficult to understand what the author means exactly with expressions such as "that transitions of the network from one region of the state space to another are plastic in nature".

7. In a similar way, some statements about results throughout the paper are made without statistical justification. As an example of this, an important demonstration of the robustness of the method would be to compare how similar is the state space across subjects. Such data is provided in Figure S4 and mentioned in the first section of the results but without any qualifications or statistics. These need to be provided. Are high and low firing rate cells mentioned as two subclasses of pyramidal cells or like the former includes fast-spiking interneurons? Both cell classes should not be merged.

8. I could not find anywhere a proper description of the single unit analysis. Even basic information such as if pyramidal cells and interneurons were separated is missing.

9. In the last section of the results Figure S12 is referred to as Figure 12 in most places.

---

## [Author Response]

Reviewer #1 (Recommendations for the authors):1. One important limitation of this method, that is not mentioned in the text, is its reliance on filtering LFPs in pre-defined frequency bands. While this is a common approach in the field, there is also numerous evidence that some oscillations, such as 'γ', do not occur in a restricted narrow band. Another approach would be to first extract oscillatory components in an unbiased manner and then construct the state space. These caveats and alternative possibilities should be discussed.

We thank the reviewer for allowing us to highlight the scope of our method. To this end, limitations, description of caveats, alternative possibilities and key advantages over traditional approaches are now added in the first and second paragraph of discussion of the manuscript.

New Additions:

While we used pre-defined frequency bands to filter oscillations from local field potentials, alternatively, unsupervised methods (see Lopes-Dos-Santos et al., 2018) can be employed to extract oscillations in an unbiased manner and construct the state space of network oscillations. We applied this framework to characterize various static and dynamic properties of the network state space during wakefulness and sleep and demonstrated multiple applications of the framework.

Traditional approaches usually consist in analyzing spectral features of individual oscillations (univariate methods) or the power and phase relationship between two oscillations (bivariate methods). This multi-variate analytical approach provides a unique window to assess the combinatorial effect of more than two oscillations on the state of the network, the neuronal population and the behavior. This provides a formal framework to uncover the potential combinations of neural signals underlying various cognitive functions during sleep and wakefulness.

2. On several occasions throughout the text causal language is used without proper justification (for example line 344). Such sentences need to be reworded carefully to distinguish what are mere correlations from actual causality.

We thank the reviewer for allowing us to modify the text. These sentences have been modified.

3. Although several analyses were conducted to illustrate potential applications of the method, in most cases is not clear what type of new information can be obtained that is also not possible to obtain with more traditional approaches. An example of this can be found in the last paragraph of page 8. Quantifying the power (and other features) of different oscillations across brain states is something that has been done multiple times employing traditional approaches. The authors need to do a better job in trying to illustrate which specific type of analysis can be done or what type of information can be extracted with his method beyond what traditional ones offer. The analysis in Figures 5 to 7 are good steps in that direction, that could be further elaborated at the expense of reducing the first, more descriptive, part of the results.

We are grateful to the reviewer for his constructive insights. We have used this direction to modify the results of the figures 4,5 and 6 have been modified to clearly describe the advantages and new information that this method provides over the traditional ones.

Additions in results of Figure 4: In particular, non-REM states in sleep2 tended to concentrate in a region of increased power in the δ and β bands, which could be the results of increased interactions with cortical activity modulated in the same range. It is also likely that such effect was induced by the exposure to relevant behavioral experience. In fact, changes in density of individual oscillations after learning have been reported using traditional analytical methods and are thought to support memory consolidation (Bakker et al., 2015; Eschenko et al., 2008, 2006). Nevertheless, while traditional methods provide information about individual components, the novel approach used here provides additional information about the combinatorial shift in the dynamics of network oscillations after learning or exploration. Thus, it provides the basis for identifying how coordinated activity among different oscillations supports memory consolidation processes, as those occurring during non-REM sleep after exploration, which cannot be elucidated using traditional analytical methods.

Additions in results of Figure 5: Γ segregation and δ decoupling offer a picture of hippocampal REM sleep as being more akin to awake locomotion (with the major difference of a stronger medium γ presence) while also suggesting a substantial independence from cortical slow oscillations. On the other hand, the across-scale coherence of non-REM sleep is consistent with this sleep stage being dominated by brain-wide collective fluctuations engaging oscillations at every range. Distinct cross frequency coupling among various individual pairs of oscillations such as theta-γ, δ-γ etc., have been already reported (Bandarabadi et al., 2019; Clemens et al., 2009; Hammer et al., 2021; Scheffzük et al., 2011). However, computing cross frequency coupling on the state space provides the additional information on how multiple oscillations, obtained from distinct CA1 hippocampal layers (stratum pyramidale, stratum radiatum and stratum lacunosum moleculare), are coupled with each other during distinct states of sleep and wakefulness. Furthermore, projecting the correlation matrices on 2D plane, provides a compact tool that allows to visualize the cross-frequency interactions among various hippocampal oscillations. Altogether, this approach reveals the complex nature of coupling dynamics occurring in hippocampus during distinct behavioral states.

Modifications in results of Figure 6: To investigate how learning during exploration affects the consequent sleep, we characterized state transition patterns during Sleep1 and Sleep2. By plotting outgoing trajectories on the network state space (Figure 6A), we visualized state transitions during sleep. We quantified the probability of state transitions among various sleep states (REM, non-REM etc.) in a transition matrix (Figure 6B). We calculated changes in transition probabilities across sleep trials by measuring average absolute change in probability (AACP, see methods). This allowed to quantify the amount of sleep state transitions altered after exploration/learning. To assess whether these changes in transition probabilities of sleep states are random or not, we generated 1000 pairs of sleep and awake trials by randomly shuffling state transitions from sleep data and computed transition matrices, difference matrix and their corresponding AACP. The AACP value of real data was then compared to the distribution of randomly shuffled trials. We observed that the AACP value of real data was beyond the 95th percentile of the distribution of shuffled data (Figure 6D), Altogether, this data highlights the specific alterations in general sleep architecture and hippocampal oscillatory landscape following learning/novel exploration.

4. Related to the previous point, a comparison that could be more interesting than REM vs non-REM is the same sleep state before and after the novel experience. While spectral changes between REM and non-REM are so obvious that can be shown with simpler analytical approaches, one could expect more subtle changes from pre to post-sleep for the same state. This could be an opportunity to showcase the power of this approach.

If we understood it correctly, the reviewer is suggesting comparing the spectral changes of ‘same’ oscillatory states across sleep trials. The ‘same’ states have identical spectral power and are thus overlapping on the state space. Alternatively, to assess the state specific alteration across sleep trials, we computed transition probabilities emerging from a given state across sleep trial. This provides an avenue to assess subtle changes in oscillatory state transition patterns after learning.

5. Results regarding Figure 5 are interesting but its description is merely qualitative. For example, in lines 223-226 some statements regarding the variable coupling of oscillations are made without any quantification/ statistics to back them up. Those should be provided either in the main text or figure legend.

Statistical quantification has been added in the Figure 5 : p < 0.001 using multivariate ANOVA (manova1 in MATLAB).

6. The language describing Figure 6 is too vague and makes it difficult to understand what the author means exactly with expressions such as "that transitions of the network from one region of the state space to another are plastic in nature".

We thank the reviewer for allowing us to clarify these findings. To this regard, the language has been modified to indicate altered sleep state transitions after exploration/ learning.

Modifications in results describing Figure 6:

Alterations in Sleep State Transitions after Exploration / Learning

To investigate how learning during exploration affects the consequent sleep, we characterized state transition patterns during Sleep1 and Sleep2. By plotting outgoing trajectories on the network state space (Figure 6A), we visualized state transitions during sleep. We quantified the probability of state transitions among various sleep states (REM, non-REM etc.) in a transition matrix (Figure 6B). We calculated changes in transition probabilities across sleep trials by measuring average absolute change in probability (AACP, see methods). This allowed to quantify the amount of sleep state transitions altered after exploration/learning. To assess whether these changes in transition probabilities of sleep states are random or not, we generated 1000 pairs of sleep and awake trials by randomly shuffling state transitions from sleep data and computed transition matrices, difference matrix and their corresponding AACP. The AACP value of real data was then compared to the distribution of randomly shuffled trials. We observed that the AACP value of real data was beyond the 95th percentile of the distribution of shuffled data (Figure 6D), Altogether, this data highlights the specific alterations in general sleep architecture and hippocampal oscillatory landscape following learning/novel exploration.

Intra-state sleep transitions are more plastic than Inter-state Transitions

We further examined which transitions on the state space are significantly altered across sleep trials. We computed AACP specifically for transition from REM/non-REM/intermediate sleep state to REM/non-REM/intermediate state. We found that transitions occurring from REM-to-REM sleep and non-REM-to-non-REM sleep (intra-state transitions) are more vulnerable to plasticity after exploration as compared to inter-state transitions (such as non-REM to REM, REM-to-intermediate etc.) (Figure 6E, F). These changes in intra-state transitions were observed to be beyond randomness (Figure S9 E, F) indicating a specificity in plastic changes in state transitions after exploration. In particular, while the average REM period duration is unaltered after exploration (Figure 4G), REM temporal structure is reorganized. In fact, increased probability of REM to REM transitions indicates a significant prolongation of REM bout duration. Similarly, the increase in non-REM to non-REM transition probability reflects an increased duration of non-REM bouts. Therefore, environment exploration was accompanied by an increased separation between REM and non-REM periods, possibly as a response to increased computational demands. More in general, the network state space allows to characterize the state transitions in hippocampus and how they are affected by novel experience or learning. By observing the state transition patterns, this analytical framework allows to detect and identify state-specific changes in the hippocampal oscillatory dynamics, beyond the possibilities offered by more traditional univariate and bivariate methods. We next investigated how fast the network flows on the state space and assessed whether the speed is uniform, or it exhibits specific region-dependent characteristics.

7. In a similar way, some statements about results throughout the paper are made without statistical justification. As an example of this, an important demonstration of the robustness of the method would be to compare how similar is the state space across subjects. Such data is provided in Figure S4 and mentioned in the first section of the results but without any qualifications or statistics. These need to be provided. Are high and low firing rate cells mentioned as two subclasses of pyramidal cells or like the former includes fast-spiking interneurons? Both cell classes should not be merged.

Comparison of data across animals is added in figure 2 —figure supplement 1. The animals exhibit high overlap on the state space.

8. I could not find anywhere a proper description of the single unit analysis. Even basic information such as if pyramidal cells and interneurons were separated is missing.

A proper description of single unit analysis has been added in the methods section of the manuscript. For this analysis / dataset, pyramidal and interneurons were not separated. All cells were included in the analysis. However, we have used hc11 datasets from Buzsaki lab with pre-classified information of pyramidal and interneurons to study sleep signatures (See Figure 8 —figure supplement 3).

9. In the last section of the results Figure S12 is referred to as Figure 12 in most places.

This has been corrected.